# Frictional healing and induced earthquakes on conventionally stable faults

Meng Li ✉, Andre R. Niemeijer ⑩ & Ylona van Dinther ⑩

Conventional studies suggest that faults in the shallow subsurface resist earthquake nucleation, because their frictional strength increases as slip accelerates (i.e., velocity-strengthening friction). Contrary to this widely held notion, such nominally stable faults frequently host earthquakes induced by human activities. Here, we resolve this contradiction using numerical models that simulate both geological and earthquake timescales using rate-and-state friction. Faults could develop significant interface strength, expressed as increase in static frictions by around 0.25, due to "healing" over thousands to millions of years. This strength gain can be released to nucleate earthquakes, also on velocity-strengthening faults. These earthquakes exhibit efficient frictional weakening similar to those natural earthquakes on velocity-weakening faults but follow revised nucleation stages and length scales. Seismic hazard for subsequent earthquakes is reduced and vastly different. Velocity-strengthening faults can no longer host earthquakes, because subsequent slip on human lifetimes is stable. Velocity-weakening fault segments may still nucleate earthquakes, but with sharply reduced stress drops. Neighboring ruptured velocity-strengthening segments impede rupture propagation and hence reduce anticipated future earthquake magnitudes. Both the increased hazard for the first induced earthquake and less hazardous subsequent events need to be properly assessed and communicated to maintain public confidence for using the subsurface for the energy transition.

Following the boom of subsurface exploitation to meet our increasing energy demand, induced earthquakes gained significant societal and scientific attention. However, these earthquakes, induced by human activities such as fossil fuel production and sustainable energy production and storage, are "unexpected" in several aspects. First, they show a distinctly different global spatial distribution from natural seismicity (Fig. 1a). Such induced earthquakes mostly take place in intraplate regions where tectonic strain rate is low or near-zero[1]. Human-induced stress perturbations are the drivers for the failure of pre-existing faults. In this way, earthquakes occur on inactive faults that lack historical seismicity and where consequently people are more at risk as infrastructure has not been built to withstand earthquakes. Moreover, these earthquakes generally occur at depths close to human activities, i.e., within the first few kilometers, which is considerably shallower than most natural earthquakes (Fig. 1b). Therefore, they can be more hazardous and cause more ground shaking[2]. In most cases, source properties of induced earthquakes, such as their moment tensor and energy spectrum, are also different from natural earthquakes[3–5]. These differences suggest that induced earthquakes might not have the same underlying physical mechanisms as natural earthquakes. The evolution of stress and strength on inactive faults has to be characterized to understand induced earthquakes and to better assess and ultimately forecast seismic hazard.

Earthquake nucleation, propagation, and arrest are widely understood to be governed by a fault's resistance to slip, which is quantified using fault friction and its velocity or displacement dependence[6–9]. Velocity-weakening (VW, $a < b$) and velocity-strengthening (VS, $a > b$) are two types of frictional properties in the

Department of Earth Sciences, Utrecht University, Utrecht, Netherlands. ✉e-mail: limeng.uni@gmail.com

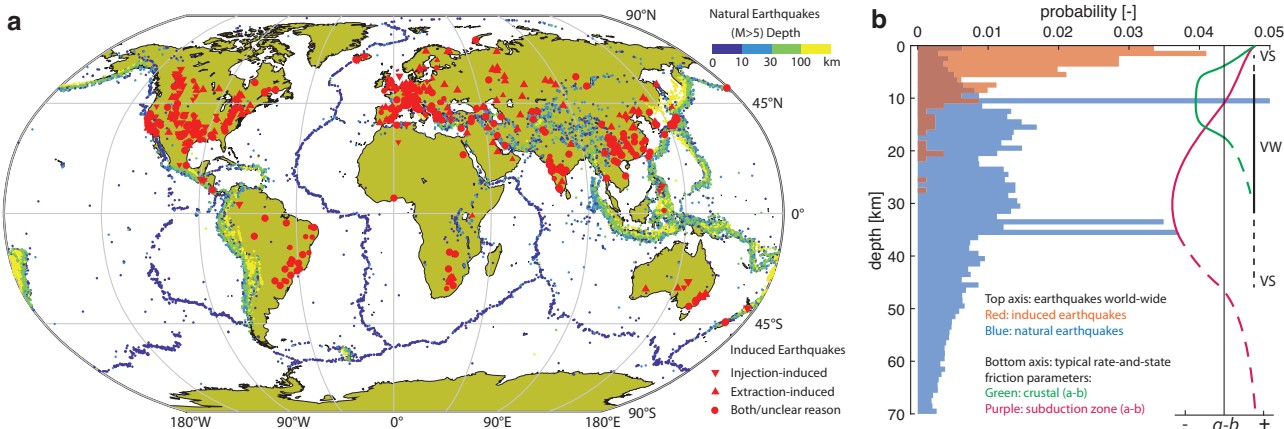

**Fig. 1 | Distribution of natural and induced earthquakes. a** Global distribution of natural and induced seismicity. Natural earthquakes with moment magnitude higher than 5 are color-coded according to their hypocenter depth (upper right colorbar, U.S. Geological Survey (USGS) catalog, year 2021)[115]. Induced earthquakes are plotted with red markers. Injection-induced, extraction-induced, and events with unclear causes are marked by different markers (bottom right legend, Human-Induced Earthquake Database (HiQuake) catalog, till 2022, note that the catalog is not exhaustive)[116,117]. Generated with MATLAB[113]. **b** Corresponding depth distribution of natural and induced seismicity. The data are from the same sources as panel **a**. The irregular peaks in natural seismicity data are due to the default depth assignment of 10, 33 and 35 km to low-accuracy earthquakes in the USGS catalog. Only the induced earthquakes reported with a depth estimate from the HiQuake catalog are plotted. The typical distributions of the rate-and-state friction parameter $(a - b)$ with depth in crustal and subduction zone settings are plotted using the bottom axis for comparison[21].

popular rate- and state-dependent friction (RSF) formulation, where $a$ and $b$ are the two frictional parameters characterizing the direct and slip-dependent response of friction coefficient to slip rate change, respectively (Fig. 1b). This single phenomenological law, originally revealed by velocity step experiments in the laboratory (with typical loading velocity ranging from 1 $\mu$m/s to 1 mm/s), provides a powerful tool to model complete earthquake sequences. During slip acceleration, VS faults become stronger while VW faults weaken, which leads to further acceleration of slip. Therefore, classical instability analysis indicates that instabilities (and thus earthquakes) only nucleate under VW friction, while VS friction favors stable sliding and thus inhibits earthquake nucleation[8,10,11]. Nucleation conditions and patterns under VW friction have been thoroughly explored with numerical simulations and laboratory experiments, where the simulated characteristics of nucleation and rupture propagation are in accordance with observations of natural earthquakes[12,13]. It is verified that earthquake nucleation includes a transition from aseismic slip to seismic slip that requires a critical length, termed as the nucleation length, to be reached[14–17]. In these studies, any VS segments on a fault function as barriers that slow down or arrest the rupture[18,19].

This concept has been questioned by several observational, experimental and numerical studies in the past decades. Numerous fault rocks at shallow depths consistently exhibit VS behavior in low temperature and pressure experiments[20–25], meanwhile induced earthquakes have been frequently reported at such depths (Fig. 1b). Evidence from the well-studied Groningen gas field in the Netherlands, the largest natural gas field in Europe and one of the largest in the world, augments this contradiction. There damaging earthquakes (up to ML 3.6) are located within or close to the depth of the gas reservoir[26–29] and are believed to be the consequence of the reactivation of pre-existing faults resulting from stress changes due to gas production[30]. Peculiarly, laboratory experiments on the simulated gouges of the reservoir rocks taken from borehole samples, show mainly VS behavior under in-situ pressure-temperature and fluid chemistry conditions[31]. This combined evidence indicates a need to clarify whether earthquakes can nucleate on VS faults or not. Both experimental and numerical studies suggest that VS faults can exhibit aseismic slip pulses after an external perturbation to the fault stress state[7,32–35]. Candidate causes of perturbation include stress concentrations caused by non-planar fault geometry, presence of bi-

material interfaces, or other external factors such as prior earthquakes and fluid presence[33,36,37].

In this work, we use numerical models to investigate the role frictional healing plays in induced earthquakes on conventionally stable faults. Despite being challenged in several studies, the concept of stable VS faults remains the convention for experts and non-experts. This enduring convention likely results from the existence of important unresolved questions. First, many studies have only simulated aseismic pulses. It has not been demonstrated if, and when, instabilities on VS faults could grow beyond aseismic slips and reach earthquake rates. Here, we show that VS faults can nucleate and propagate earthquakes, but only once; a recurrent sequence is not possible on relevant time scales. Second, the origin of the stress perturbations proposed to trigger instabilities has been poorly identified or justified in observations. Instead of introducing additional factors, we hypothesize that an instability can emerge from frictional healing – a mechanism inherently active on almost every fault. That means that earthquakes can nucleate even on planar homogeneous VS faults devoid of external perturbations, and hence the proposed external perturbations are not a necessary requirement. Fault healing, as previously observed in laboratory experiments, is expected to become important under long-term tectonic inactivity. However, fault healing over geological timescales and its impact on earthquake sequences have not yet been simulated. Third, there is an absence of studies aiming to understand the presence and characteristics of nucleation and rupture behavior of induced earthquakes under VS friction. We quantify these features, including the characteristic nucleation lengths, using both numerical and theoretical approaches. These characteristics and quantification provide the way forward for inclusion into earthquake studies, seismic hazard assessment, and ultimately to facilitate our ability to use the subsurface to benefit the energy transition.

## Results

### Healing: from laboratory to tectonic timescales

Fault healing is important under long-term tectonic inactivity and thus could be key to explain why areas devoid of natural earthquakes may still be prone to seismicity in induced scenarios. Fault healing describes the time-dependent recovery of fault strength observed in slide-hold-slide experiments, where the peak stress of the subsequent

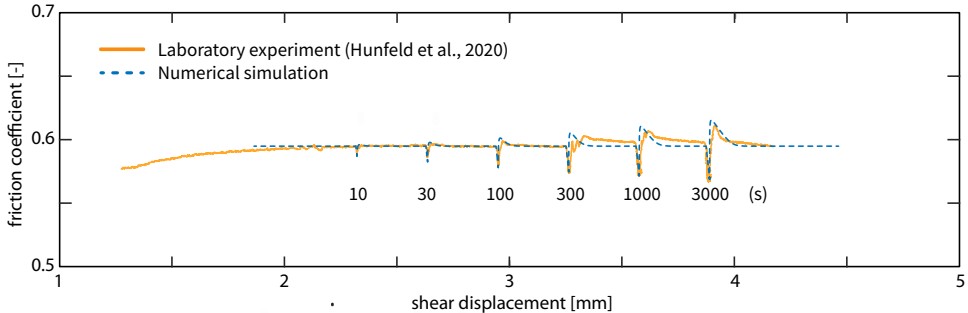

**Fig. 2 | Comparison of numerical simulation and laboratory healing of different durations.** Laboratory data are from the experiment UU-SS-01 using Groningen Slochteren sandstone[46], numerical results are from the 0-D model in this study. Healing durations of 10 to 3000 s were tested.

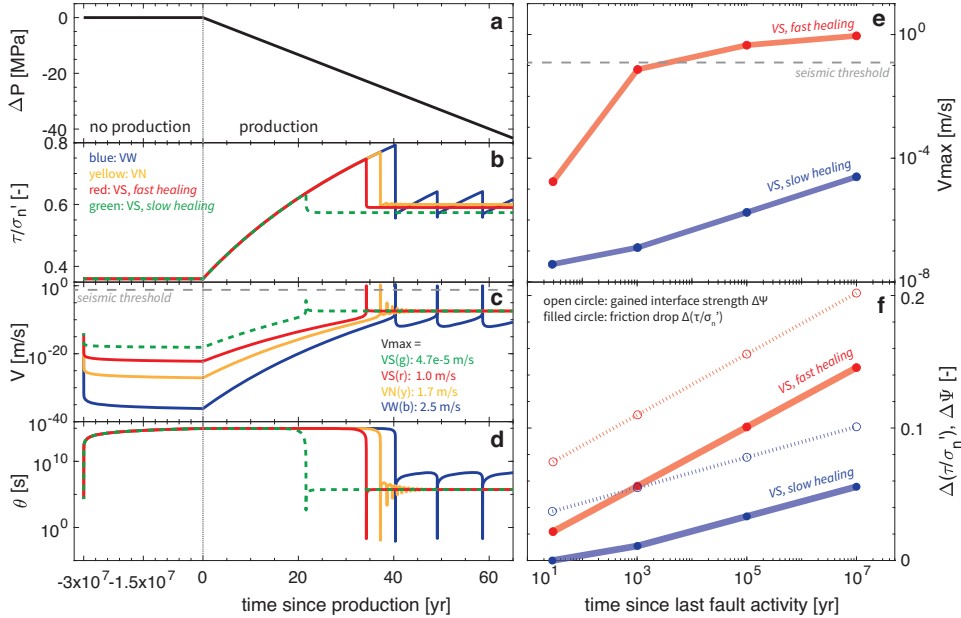

**Fig. 3 | Earthquake sequences after healing over geological time scales.**
**a**–**d** Simulation of earthquake sequences in 0-D in velocity-strengthening (VS) (red: $a = 0.013$, $b = 0.01$; green: $a = 0.013$, $b = 0.005$), velocity-weakening (VW) (blue: $a = 0.007$, $b = 0.01$) and velocity-neutral (VN) (yellow: $a = 0.01$, $b = 0.01$) scenarios with a healing time of 30 million yr, with fixed $D_{RS} = 2$ mm. **a** The pore pressure, **b** the ratio of shear stress and effective normal stress $\tau/\sigma'_n$, **c** the slip rate, and **d** the state variable are plotted with respect to time since production starts. Negative time refers to the time before production. Different scales are used in the positive and negative parts of the axis. **e**, **f** Simulation of earthquake sequences in VS scenarios ($a = 0.013$) with different lengths of healing time from 10 yr to 10 million yr. **e** The maximum slip rate $V_{max}$ of the induced events is plotted with respect to the time of the event since the last fault activity, i.e., where healing starts. **f** The friction drop $\Delta(\tau/\sigma'_n)$ (i.e., change in the ratio of shear stress to effective normal stress) and the gained interface strength $\Delta\Psi$ during healing are plotted with respect to the time of the event since the last fault activity.

slip ("slide") after fault reactivation increases with the time of inactivity ("hold")[38–43]. In this way, larger stress drops are obtained after a longer healing time. The observed strength increase originates from the growth of grain contacts (asperities) and compaction via pressure solution[44] as well as the development of cohesion[45]. Experimentally healed fault rocks show evidence of cementation and surface energy driven grain boundary healing at microscopic scale[46]. This behavior is confirmed also for the specific VS lithologies in Groningen[46]. It has been suggested that the healing rate is proportional to the RSF parameter $b$[47]. Large $b$ indicates fast healing and vice versa. Our numerical model quantitatively reproduces the healing behavior in slide-hold-slide experiments on simulated fault rocks of the Groningen reservoir and the aseismic slip events (i.e., slips slower than a seismic threshold) after laboratory timescales of healing (10-3000 s, Fig. 2). Given that the Groningen region and many other sites of human activities have been tectonically inactive for millions of years[48], we will evaluate whether earthquakes (i.e., fast fault slips) can nucleate on VS faults after healing over geological timescales.

We extend the healing time from laboratory scales to hundreds of millions of years to introduce a numerical model that resolves fault slip on both geological and earthquake time scales (Fig. 3). We simulate a typical gas production site configuration in 0-D and 2-D (Fig. 4, see also Methods and parameters in Supplementary Table 1). Our 0-D model simulates the top inner corner of the reservoir, where the fault is most critical upon gas production. For comparison, we model VS and VW faults, with high and low healing rates, and various healing periods.

We find that induced earthquakes can nucleate on VS faults, similar to their VW counterparts (solid lines in Fig. 3b–d). Both types of faults first experience millions of years of inactivity during which their stress level remains low and far from failure. This simulates the period after the last fault activity ("no production" period in Fig. 3a, b). The slip rate rapidly decays to practically zero, which matches observations in the laboratory[47] (Fig. 3c). After this period of quiescence, we prescribe a sudden, continuous pore pressure change simulating human activities such as gas production ("production" period in Fig. 3a). During reservoir depletion, both shear stress and effective normal

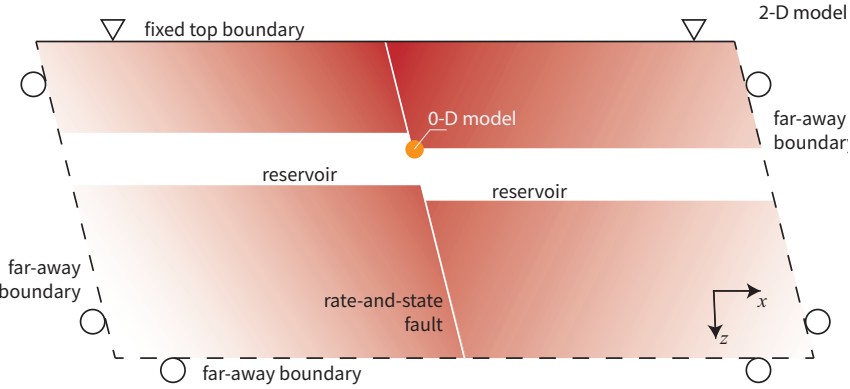

**Fig. 4 | Model setup of a normal fault crosscutting a depleting gas reservoir.** The setup is modeled in 2-D with plane-strain assumption. The simplified 0-D model, in which only the top inner corner of the reservoir is modeled, is marked in orange. See Methods for the explanations of the boundary conditions.

stress on the fault increase with time. Because of the rock's low Poisson ratio, the shear stress increases faster than the effective normal stress due to the poro-elastic effect[49], pushing the fault towards failure (Fig. 3b). Fault strength (i.e., the maximum static friction) is reached after a certain period of time, which is the aseismic period between the start of human perturbation and the start of induced seismicity, as observed both in the field[50] and in the laboratory[46,47]. As on VW faults, VS faults can accelerate to seismic slip rates (cm/s to m/s) after failure, generating an earthquake. In this simulation tuned to the Groningen configuration with a healing time of 30 Ma[51], the first earthquake with a maximum slip rate of ~1.0 m/s occurs about 35 years after the onset of gas production. This aseismic period is comparable to the observed aseismic period in Groningen (1963–1991)[50]. The simulated stress drop is about 3.0 MPa, which is similar to the 2.5 MPa reported for the 2012 Huizinge ML 3.6 earthquake[52].

After the first earthquake, slip on VS faults is stable, if human activities continue (Fig. 3b–d). The simulated VS fault reaches a steady state in which it moves at a slow constant slip rate defined by the rate of human activity. Since the ruptured VS fault segment is not re-locked on human timescales, that segment cannot nucleate future earthquakes, nor can it be ruptured by neighboring ruptures. In contrast, recurring earthquake sequences follow on VW faults. However, subsequent slip rates and stress drops never reach the magnitude of the first earthquake, because fault strength does not recover to a healed friction of 0.8. Maximum friction rather remains at ~0.65, which is the typical static friction without healing, thus reducing the potential stress drop by a factor 2 or more. Instead of earthquakes, several slow slip events follow when the fault is close to velocity-neutral (VN, $a = b$). In all these scenarios, the fault becomes less hazardous after the first earthquake, because the added seismic potential due to the healed strength has been released.

If the healing rate is low, as represented by a RSF parameter $b$ of 0.005 instead of 0.01, a VS fault experiences slow slip events rather than earthquakes (dashed green line in Fig. 3a–d). This fault essentially has experienced a similar healing history during the non-production period, as reflected by the same amount of logarithmic increase in the "state variable", which is often interpreted as the average age of the asperity contacts (Fig. 3d). However, this fault has been less locked, as indicated by its higher slip rate prevalent for millions of years (Fig. 3b). Hence, the fault has healed less and fails at a lower strength after a shorter period of production (20 years). Failure occurs in the form of a slow slip event with slip rates that only reach $10^{-5}$ m/s and a stress drop that is one order of magnitude smaller. These results support the interpretation that the RSF parameter $b$ reflects the rate of healing. A low healing rate might thus be an explanation for induced aseismic events.

Besides healing rate, healing time is another factor that contributes to earthquake occurrence on VS faults. To vary the healing time, we

activate the production at different times since the last fault activity (Fig. 3e–f). For both fast and slow healing rates, higher slip rates are recorded when healing time increases (Fig. 3e). Specifically, the maximum slip rate grows linearly with applied healing time on slowly healing VS faults. The growth becomes logarithmic on the fast healing fault because the wave-radiated energy loss could suppress slip rates[53]. In our simulations, the fast healing VS fault ($b = 0.01$) becomes seismogenic after healing over 100,000 years, while the slowly healing VS fault ($b = 0.005$) remains aseismic even after healing for 30 Ma (Fig. 3e). Therefore, both a long healing period and a high healing rate are prerequisites for the nucleation of earthquake ruptures on VS faults.

The increase of fault strength due to healing is better quantified with the introduction of "interface strength". The growth of breakdown stress drop, expressed in the difference between maximum and minimum friction coefficients, with healing time is logarithmic on both fast and slowly healing faults, but at different rates (Fig. 3f). This increase in stress drop comes from an increasing fault strength with respect to an unchanging dynamic friction. The concept of interface strength

$$\Psi = \mu_0 + b \ln\left(\frac{\theta V_0}{D_{RS}}\right) \qquad (1)$$

was introduced by ref. 47 (see also Methods). It becomes clear that slip rate $V$ is defined by the relative amplitude of the applied shear force $\mu = \frac{\tau_s}{\sigma'_n}$ with respect to $\Psi$ when RSF is reformulated as

$$\frac{V}{V_0} = \exp\left(\frac{\mu - \Psi}{a}\right). \qquad (2)$$

This equation illustrates that a high slip rate is achieved when interface strength is overcome ($\mu > \Psi$), simulating the coseismic phase. In contrast, a locked phase is simulated if the shear force is below the interface strength ($\mu < \Psi$). In this way, the interface strength can be used to represent the classical concept of fault strength. In the healing phase with a near-zero slip rate, RSF predicts a linear growth of the state variable with time ($\Delta\theta = \theta - \theta_i = t$) (Fig. 3d) and thus a logarithmic growth of the interface strength [$\Delta\Psi = \Psi - \Psi_i = b \ln(1 + t/\theta_i)$], where $\Psi_i$ and $\theta_i$ are the initial values of interface strength and state. In our simulations, the increased interface strength $\Delta\Psi$ with healing time as predicted by the equation is exactly the same amount as the increased stress drop $\Delta(\tau_s/\sigma'_n)$, with the increasing rate equal to $b$ (Fig. 3f, see also the evolution of friction and interface strength in Supplementary Fig. 1). Thus $b$ can accordingly be called the "healing rate"[41,47]. Note that the interface strength is different from the traditionally defined fault strength by a constant that comes from the arbitrary choice of $V_0$ and $\mu_0$ in RSF. Hence, only the interface strength growth is equivalent to the fault strength growth. After 30 million years of healing, the

interface strength of our fast healing fault tuned for Groningen increases by 0.23, such that fault strength at failure is approximately 0.8. A corresponding strength increase of about 4 MPa agrees with the Coulomb stress threshold inverted from statistical seismicity rate models[54].

In summary, VS faults can nucleate single earthquakes under human activities similar to VW faults, whereas subsequent slip is stable. Earthquake occurrence on VS faults requires fault healing over geological timescales at a high healing rate, such that fault strength can be elevated enough. Faults that heal slowly or for insufficient time remain aseismic.

### Earthquake nucleation on velocity-strengthening faults

The nucleation and subsequent rupture behavior of induced earthquakes on VS faults have not been studied. Whether a fault is VS or VW also affects the spatial evolution of slip on the fault in the period between fault reactivation and seismic rupture, i.e., during the nucleation phase. We use 2-D models and theory to quantify nucleation phase characteristics, such as the characteristic nucleation lengths. In a 2-D model representing a Groningen fault (Fig. 4), the shallowest point of the fault inside the reservoir reaches the failure stress first, after the fault has been re-activated due to gas production (Fig. 5a). The nucleation process starts here and expands downwards to occupy more of the depth interval of the gas reservoir. The whole process can be separated into three stages according to the slip rate reached at the nucleation front: (i) fault activation, when the maximum slip rate rises from near-zero to a background rate (controlled by the pressure depletion rate); (ii) nucleation, when the maximum slip rate is between the background rate and the seismic threshold determined by the dominance of wave radiation (Equation (9))[11]; and (iii) rupture propagation, when the maximum slip rate is above the seismic threshold.

Nucleation of induced earthquakes on VS faults is comparable to that on VW faults during all three stages, without a sharp contrast at the VS-VW transition. We compare the nucleation pattern on VW and VS faults by gradually increasing the value of $a/b$, while keeping the healing rate $b$ and healing time fixed. We find that stage (i) does not change with $a/b$. The nucleation zone reaches the same length of $L_I$ after this stage (Fig. 5a, c, e, also refer to Supplementary Fig. 2). However, the nucleation length before stage (iii), named $L_{III}$, varies with $a/b$. For $a/b \leq 0.5$, the nucleation process quickly completes at a small $L_{III}$, leaving the remainder of the fault to be ruptured seismically ($a/b = 0.5$ in Supplementary Fig. 2c). $L_{III}$ increases as $a/b$ increases ($a/b = 0.7$ in Fig. 5e). When $a/b$ is close to 1, nucleation terminates at the fault boundary without reaching seismic threshold due to the finite fault size, leaving the stage (iii) unobserved ($a/b = 0.9$ in Fig. 5c). A similar pattern of the expansion of the nucleation zone is found on VS faults ($a/b = 1.1$ in Fig. 5a, $a/b = 1.3 \& 1.5$ in Supplementary Fig. 2a–b). There is no distinct change at $a = b$, which has conventionally been assumed to be the boundary between instability and stable creep (Fig. 5a vs. c). The nucleation behavior seems to transition smoothly when $a/b$ is increased above 1.

The analogy of nucleation and rupture behaviors between VS and VW faults reflects a shared mechanism that allows slip acceleration on both faults: slip-weakening friction, which is inherent in RSF as long as $b$ is positive. Independent of the $a/b$ ratio, the stress evolution follows the same pattern after failure: shear stress drops from the fault strength to a dynamic stress almost linearly with a similar weakening slope (or rate) $k_{sw} = -b/D_{RS}$, until a similar amount of slip, usually referred to as the slip-weakening distance $D_{sw}$, is reached (Fig. 5b). Different locations on the fault follow approximately the same pattern, with similar stress drop and slip-weakening distance (semi-transparent lines in Fig. 5b, d, f, also Supplementary Fig. 2). Since the area of the triangle below the slip-weakening curve defines the fracture energy $G_c$[55] (Fig. 5b), all these VS and VW faults have approximately the same

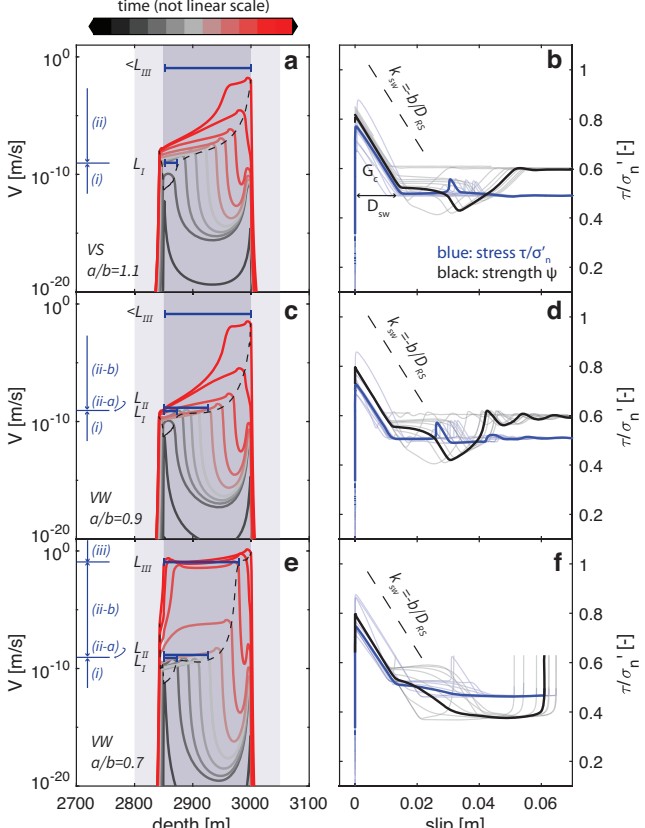

**Fig. 5 | Simulation of the first induced earthquake in 2-D in VS and VW scenarios with several $a/b$ ratios. a**, **b** VS, $a/b = 1.1$, **c**, **d** VW, $a/b = 0.9$, **e**, **f** VW, $a/b = 0.7$ with fixed $b = 0.01$ and $D_{RS} = 0.5$ mm, $\mu_0 = 0.5$, healing time $t_h = 100$ million yr. **a**, **c**, **e** The temporal evolution of slip rate. The interseismic and nucleation phases are plotted in black and gray, with gradual transition to the coseismic phase in red (colorbar above). The plotted lines are not in regular time intervals. The nucleation stages (i)–(iii) are indicated by blue arrows next to the vertical axes, separated by the maximum slip rate achieved (stage iii only exists in **e**). The measured nucleation lengths $L_I$, $L_{II}$ and $L_{III}$ are shown as blue bars on the top ($L_{II}$ only measured in **e**). The dashed black lines track the temporal-spatial evolution of the nucleation front, characterized as the propagation of the local stress peaks. The blue shadows in the background specify the range of the reservoir depth on the hanging wall side (2850–3050 m) and the footwall side (2800–3000 m). The detailed model setup is depicted in Fig. 4. **b**, **d**, **f** The evolution of friction $\tau/\sigma'_n$ with respect to slip (blue). The evolution of interface strength $\Psi$ (Equation (1)) is plotted in black for reference. The multiple semi-transparent lines in the background are the observations from different locations (every 20 m between 2800 m and 3050 m depth) on the fault. The bold lines are the observation at the center of the reservoir (2950 m depth). The dashed line indicates the slip-weakening slope of $k_{sw} = -b/D_{RS}$ for reference. The slip-weakening distance $D_{sw}$ and the fracture energy $G_c$ are labeled in panel **b**.

amount of fracture energy, explaining the shared nucleation front dynamics. Knowing that $G_c$ is proportional to $b$ (Methods)[11], fast healing faults always have high $G_c$, which is commonly linked to large earthquake magnitudes via observational and numerical studies[56]. We find that these relevant mechanisms, previously revealed exclusively for VW friction[57,58], can be extended to VS friction as well. Therefore VS faults can experience seismic energy release in the same manner as VW faults.

Nevertheless, the nucleation stage (ii) on VS faults exhibits the most dissimilarity to VW faults, where nucleation length $L_{II}$ is eliminated. To demonstrate this, we collect the spatial evolution patterns of the nucleation front under different $a/b$ ratios between 0.5 and 1.5 (black dashed lines in Fig. 5a, c, e and Supplementary Fig. 2) and plot them in the same figure (Fig. 6a). We find that the lines overlap at stage

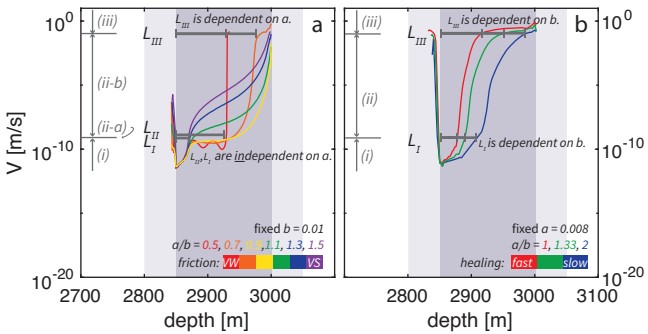

**Fig. 6 | Characteristic nucleation lengths and their dependence on *a/b*. a** The temporal-spatial propagation of the nucleation front extracted from Fig. 5 and Supplementary Fig. 2. The nucleation stages (i)-(iii) are indicated by gray arrows next to the vertical axes, separated by the maximum slip rate achieved. The measured nucleation lengths $L_I$, $L_{II}$ and $L_{III}$ are shown as gray bars on the top ($L_{II}$ and $L_{III}$ only measured in some scenarios). **b** The temporal-spatial propagation of the nucleation front in another set of experiments with fixed $\mu_0 = 0.3$, $a = 0.008$ and varied $b = 0.008, 0.006, 0.004$, see Supplementary Fig. 4.

(i), but start to diverge during stage (ii). For VW faults, this stage can be separated into two sections (ii-a) and (ii-b). The nucleation zone first expands its length from $L_I$ to $L_{II}$ slowly but steadily in section (ii-a). During this section, the slip rate at the nucleation front remains at the background rate. The nucleation speed also remains approximately constant (Supplementary Fig. 3a). This section (ii-a) to reach the same $L_{II}$ that is independent on $a/b$ is on all three VW faults with $a/b$ between 0.5 and 0.9. The nucleation front then accelerates quickly to sub-Rayleigh speeds together with its expansion to $L_{III}$ in section (ii-b). Because an $a/b$-dependent $L_{III}$ needs to be reached at the end of this section (ii-b), the nucleation fronts start to follow different paths during this section. However, nucleation fronts start to diverge from the beginning of stage (ii) on VS faults, with the section (ii-a) of VW faults skipped. The transitional length scale $L_{II}$ is not observed on VS faults (Fig. 6a). Immediate acceleration and expansion of the nucleation region is seen as soon as $L_I$ is reached. The nucleation front keeps accelerating during stage (ii) (Supplementary Fig. 3b). The acceleration is fast at the beginning, then slows down, and speeds up again. Contrary to intuition, more VS faults usually have higher slip rates and nucleation speeds during stage (ii) compared to less VS faults. In summary, we find nucleation on VS faults share two nucleation lengths $L_I$ and $L_{III}$ with VW faults, whereas $L_{II}$ is eliminated. We use the slip rate at the nucleation front to distinguish different nucleation stages. The same nucleation stage separation and nucleation pattern are also reflected in the spatial evolution of the nucleation speed (Supplementary Fig. 3c), since slip rate is proportional to the propagation speed of the nucleation front[59].

The three identified length scales either follow previously discovered length scales for VW faults or have to be revised for induced earthquakes. We compare them and use their dependence on $a/b$ to evaluate the applicability of various analytical expressions of nucleation length scales derived in previous studies for VW faults[11,60,61]. The cohesive zone length $\Lambda_0 = \frac{9\pi}{32}\frac{GD_{RS}}{b\sigma(1-\nu)}$[61], also known as the process zone length, matches our simulated length scale $L_I$. It is the smallest length scale during nucleation, and appears to apply to both VW and VS faults (Fig. 6). This length scale is independent of $a$ (Fig. 6a). Its proportionality to $b$ holds for VS faults as well, as supported by our simulations where healing rate $b$ is varied (Fig. 6b). Another nucleation length $2L_b = 2\frac{GD_{RS}}{b\sigma(1-\nu)}$[11] matches the simulated length scale $L_{II}$ on VW faults. This length scale is independent of $a/b$ (Fig. 6a), but it does not appear to be a critical length scale on the VS faults we simulated, no matter if we vary $a$ or $b$ (Fig. 6a, b). In fact, this length scale (or multiplied by a geometrical constant of about 1), sometimes also referred

to as the minimum nucleation length, is not always present even on VW faults according to previous numerical models[11,62–64]. Whether it can be simulated or not relies on the initial and loading conditions. Thus, the applicability of this length scale to VS faults needs further exploration under different conditions. A third suggested nucleation length $2L_c = \frac{2}{\pi}(\frac{b}{b-a})^2 L_b$[11] compares with our simulated $L_{III}$. However, the expression of $2L_c$ has to be revised because it is not meaningful when $a \geq b$ despite the squared term $(\frac{b}{b-a})^2$. This is because a stress drop term utilized to drive the expression of $2L_c$ is proportional to $(b-a)$ and must be positive (see Methods). This requirement reflects why natural earthquakes do not happen on VS faults, whereas induced earthquakes have elevated fault strength and hence increased stress drop. We take the healing-induced strength increase into account to obtain the expression of $L_{III}$ for induced earthquakes. The revised expression (Equation (23)) is valid for healed faults, either VS or VW. $L_{III}$ keeps the dependence and trend of $2L_c$ on both $a$ and $b$: it increases with increasing $a$ (Fig. 6a) and decreasing $b$ (Fig. 6b).

In summary, nucleation of induced earthquakes on VS faults is comparable to that on VW faults, but with unique features. Nucleation on both types of faults starts with the same fault activation stage (i) regulated by length scale $L_I$, and ends at length scale $L_{III}$, which needs to be revised for induced earthquakes, before rupture propagation stage (iii). The major dissimilarity is during nucleation stage (ii) where a steady nucleation expansion section (ii-a) and consequently a transitional length scale $L_{II}$ are eliminated for VS faults. The similarity of nucleation and rupture behaviors on VS and VW faults reflects the shared mechanism of slip-weakening friction that controls the stress drop and fracture energy. This mechanism is inherent in RSF as long as $b$ is positive. Fast healing faults have large $b$ and therefore have more efficient slip weakening and large fracture energy at the same time. They could potentially nucleate larger earthquakes.

## Seismic potential of faults in reservoir rocks

We study the influence of frictional parameters on the size of the induced earthquakes and associated seismic hazard. We use stress drop as a proxy for earthquake magnitude to avoid additional assumptions to empirically upscale our simulation results to 3-D. We systematically explore the parameter space of RSF parameters $a$ and $b$, characteristic length $D_{RS}$ and reference friction $\mu_0$ within the range indicated by laboratory experiments (Supplementary Table 1, except for $D_{RS}$ which is 1-2 orders of magnitude larger, as suggested[65]). We find that for both VS and VW faults, a longer healing time increases the simulated stress drop of the first induced event (Fig. 7, also seen in Fig. 3e, f). The simulated stress drop also increases if the fault is less VS (also seen in Fig. 3b, c). Specifically, we find that the scattered stress drops collapse onto a function of the product of a healing term $b \ln t_h/\theta_{SS}$ (where $\theta_{SS}$ is the state variable at steady state after the earthquake) and the ratio $b/a$ (Fig. 7). The stress drop appears to be independent of $D_{RS}$ and $\mu_0$ in 0-D simulations.

We further use the slip rate as a proxy for ground motion and thus a metric for seismic hazard, as a larger energy release from the source usually translates to larger ground motions[66]. In hazard assessment, it is common to assess the probability of the occurrence of an earthquake of a certain magnitude to evaluate the hazard. Similarly, we quantify how much healing time would be required to make the first induced earthquake reach slip rates of 1 m/s under different frictional properties (Fig. 8, Supplementary Fig. 5). We find that a large variety of VS faults ($a/b < 1.3$) are capable of generating earthquakes with >1 m/s slip rates after healing over geological timescales (~100 Ma), as long as they have moderate to high healing rates ($b > 0.005$). This range is extended further to $a/b < 1.7$ and $b > 0$ when we also consider enhanced dynamic weakening (DW) mechanisms, which might be activated due to frictional heating when slip rates exceed 1 cm/s. Several DW mechanisms have been proposed and we included flash heating in our model, as this theory is well-supported and easy to

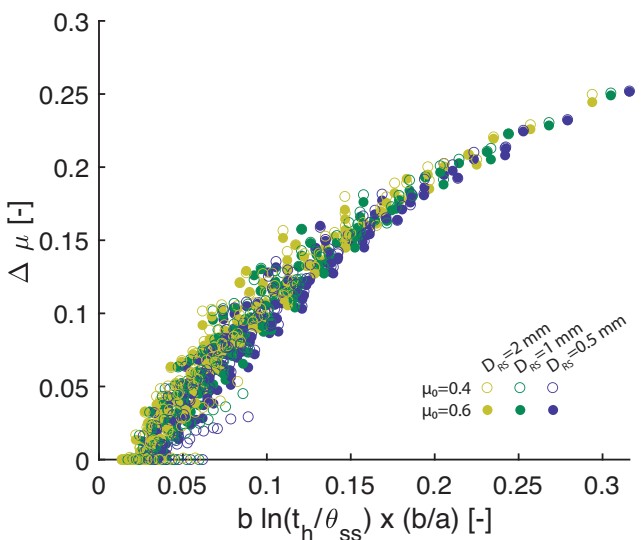

**Fig. 7 | Parameter study of the impact of frictional properties and healing time on earthquake stress drop.** The friction drop $\Delta\mu = \Delta(\tau/\sigma'_n)$ as a function of healing time $t_h$ and $a/b$, in 0-D simulations with other parameters such as static friction $\mu_0$ and characteristic length $D_{RS}$ also varying. The horizontal axis is the product of the effect of healing [$b\ln(t_h/\theta_{ss})$] and the effect of frictional property ($b/a$), where $\theta_{ss}$ is the state variable at steady state after the earthquake.

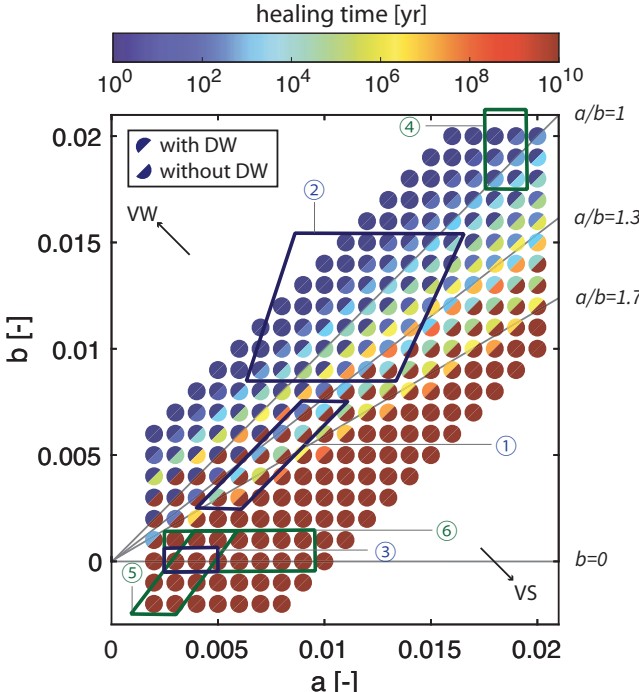

**Fig. 8 | Parameter study on seismic potential.** The required healing time for the first induced earthquake to reach a slip rate of 1 m/s. The simulation results without dynamic weakening (DW) mechanism taken into consideration are shown by the color of the bottom right half of the circle, and the results with DW are shown by the color of the top left half of the circle. The blue quadrilateral labeled as "1" encloses the parameter range of the typical Groningen reservoir rock, the Slochteren sandstone. The other quadrilaterals enclose the parameter range of the typical lithologies above or below Groningen reservoirs (blue series labeled 1-3) and other reservoir rocks (green series labeled 4-6) documented in the laboratory experiments. Labels refer to the following experiments. Blue: Groningen lithologies under $\sigma'_n = 40$ MPa, 100 °C, wet condition[31,46]; 1: Slochteren sandstone; 2: Basal Zechstein; 3: Ten Boer claystone and Carboniferous shale (same range). Green: other reservoir rocks under room temperature and wet condition; 4: carbonate from Monte Maggio (Italy) under $\sigma'_n = 10$ MPa[118]; 5: clay-rich shale from Rochester (U.S.) under $\sigma'_n = 100$ MPa[23]; 6: basalt from Xiashan landslide (China) under $\sigma'_n = 5$ MPa[25].

use[67,68] (see Methods). Flash heating further reduces the dynamic frictional strength and thus amplifies the stress drop and slip rates. Consequently, a shorter healing time is needed before an earthquake identified by a slip rate threshold could occur (Fig. 8).

We use the quantification in Fig. 8 to "forecast" the occurrence of future induced earthquakes in a region, if good constraints on the fault properties and time since last fault activity are available. Given that healing is logarithmic with time, a fault's seismic potential will hardly change on human timescales. Thus slowly healing faults that are currently non-seismogenic would not turn seismogenic within the concerned human timescales. Our forecast is exemplified through labeling the measured frictional properties of representative reservoir rock types in Fig. 8. Faults in the reservoir rocks hosting earthquakes in Groningen (Slochteren sandstone)[31] need a healing time of 25 Ma at most before an earthquake with slip rate higher than 1 m/s can occur, when enhanced dynamic weakening is considered (blue label 1 in Fig. 8). This explains the observed seismicity, since the last fault activities occurred ca. 30-65 Ma ago according to geological studies[51,69]. On top of the Slochteren sandstone layer, there is a layer of Basal Zechstein caprock (mostly anhydrite) that is weakly VW under specific salinity conditions[31], indicating its seismic potential, with or without long-term healing (blue label 2 in Fig. 8). In contrast, the Ten Boer claystone and Carboniferous shale layers around the reservoir are VS with negligible or even negative healing[46], suggesting their inability of hosting nucleation independent of healing time (blue label 3 in Fig. 8). We find that this inference is in line with seismological inversions, which have located the earthquake hypocenters either inside the sandstone reservoir or in the Basel Zechstein caprock layer[27–29]. Which layer will eventually nucleate an earthquake will also depend on whether its thickness fulfills the nucleation length threshold $L_{III}$. In more realistic but complicated scenarios (like Groningen) where several rock layers of mixed frictional properties coexist in the seismogenic zone, nucleation starting from one layer may extend to its surrounding layers, if the original rock layer cannot accommodate the full nucleation process. The induced slip rate will then depend on the healing history and slip-weakening behavior of all layers the nucleation front goes through.

Besides sandstone, we extend our inference on seismic potential to other representative reservoir rocks across the globe by only evaluating their frictional properties. Carbonate rocks (limestone, dolomite), constituting half of the known conventional oil reserves and hosting a great number of natural and induced earthquakes[70–72], often show VW or weakly VS behavior with high healing rates in experimental studies[43,73] (green label 4 in Fig. 8). Such rocks are thus potentially seismogenic based on classical instability analysis and healing will further promote earthquake occurrence. As a representative of unconventional reservoirs, shales are hydraulically fractured for "shale gas", commonly in North America, and are believed to attribute to the largest induced earthquakes in many regions[74–76]. Basalts are explored as a host rock for alternative CO2 storage solution, based on their mineral carbonation ability[77]. Laboratory studies on both shales and basalts report a wide range of frictional properties and healing rates. Some of them are measured as VS with a near-zero or sometimes negative healing rate, such as clay-rich or phyllosilicate-rich shales[23,78,79] and some basalts[25,80] (green labels 5 and 6 in Fig. 8). Although shales with low static friction ( ~ 0.3 in wet condition) might be activated by fast pore pressure change[35], it is yet to be clarified under which conditions such instability will grow to an earthquake or will remain aseismic as proposed by[81,82]. Also basalts have been reported to be VW and/or have high healing rates in some studies,

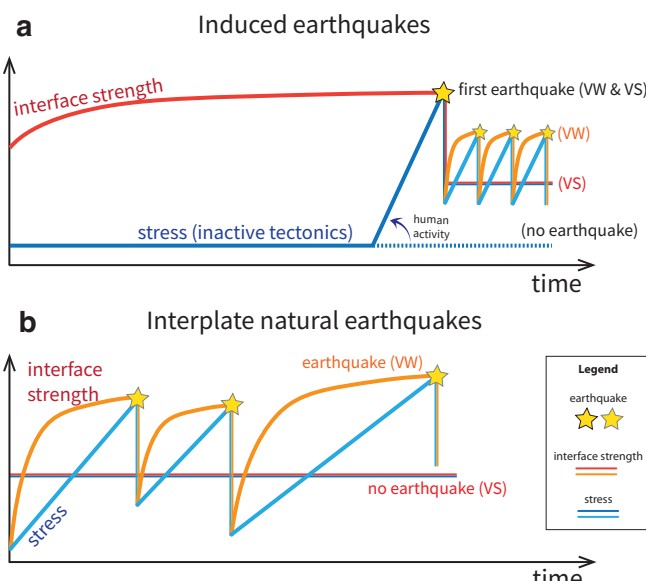

**Fig. 9 | Mechanisms of natural and induced earthquakes.** Schematic explanation of stress (blue colors) and interface strength (red colors) evolution for induced and natural earthquakes. **a** Schematic comparison between the induced earthquake sequences on VW and VS faults. The dotted lines are continued stress evolution if without human activities. **b** Schematic comparison between the interplate natural earthquake sequences on VW and VS faults, inspired by ref. 86.

depending on the temperature and composition[25,83], increasing their seismic potential. Since $CO_2$ storage involves chemical reactions that change the mineral composition, precipitation of carbonates might enhance the seismogenic potential due to its fast healing, but this has not been investigated yet[84].

## Discussion

Induced earthquakes regularly shake tectonically inactive regions, which are not prepared for such shaking. These earthquakes would not occur without human activities, irrespective of frictional stability ($a/b$ ratio) of the fault (Fig. 9a). Potential stress drop comes from the elevated fault strength, which can reach up to 0.7-1.0 after healing over millions of years (Figs. 3, 5, 7). This suggests that static friction and fault strength might be notably higher than typically used in earthquake models and hazard assessment ( ~ 0.6). Under the same stress conditions, this makes healed faults less critical than currently thought. The non-criticality of faults is evidenced by the aseismic period between the start of human activities and the first recorded event. In Groningen, this period is about 30 years, during which pore pressure had to be reduced by tens of MPa to make some faults critically stressed[85]. Admitting larger fault strengths will also lead to a knock-on effect when inferring other parameters and conditions from observations. For example, earthquake models use and tune the background tectonic stress as an initial condition to simulate the characteristics of observed seismicity. To still match the 30-year aseismic period as observed in Groningen, such models would need, and hence infer, either different initial stress conditions and/or other material parameters to obtain faster loading[54].

Notably, the subsequent fault behavior after the first induced earthquake is less devastating after the healed strength has been removed. Subsequent hazard is downgraded to what conventional stability theory predicts: constant creep on VS faults, and recurring earthquakes on VW faults (Figs. 3, 9a). For VS faults, earthquakes will not return on human lifetimes and subsequent slip is stable (Fig. 9a). This is the same as the steady-state behavior on natural VS fault behavior under tectonic loading (Fig. 9b)[86]. Ruptured VS fault

segments will not be re-locked under continuous anthropogenic loading. They cannot nucleate future earthquakes on human time-scales, nor can they be ruptured by neighboring seismic events. Such faults do re-lock and re-heal if human activities discontinue. However, the fault still cannot re-gain the same amount of strength on human timescales. Thus an earthquake of the same size as the first induced earthquake will not recur, either if the human activities will continue or be reactivated.

For VW faults, subsequent earthquakes will recur, but with sharply reduced potential stress drops (Fig. 9a). This recurrence is also similar to natural VW fault behavior under tectonic loading (Fig. 9b), except that the recurrence interval is short, since human-induced stress changes are often incomparably faster than tectonic loading. Fault strength at the occurrence of these earthquakes is substantially reduced compared to the first induced earthquake. It cannot recover to the level before human activities during the short recurrence interval. Instead, fault strength at failure remains at ~ 0.65, which is approximately the static friction of fault rocks without healing. Hence, it is thus no longer necessary to consider interseismic healing for the subsequent induced events on VW faults. It is essentially for the same reason as why healing has been ignored in natural interplate earthquake sequences: fast coseismic and postseismic healing mechanisms will be more decisive than the interseismic healing for such short recurrence intervals[87,88].

Therefore, for induced seismicity in a complex fault network with a mixture of VW and VS faults, we still expect the devastating ones to be the first induced earthquakes on each healed fault, instead of in the subsequent earthquake sequences. Moreover, rupture of first earthquakes could trigger the activation of neighboring faults that have not been ruptured, escalating seismic hazard. In contrast, subsequent ruptures on VW faults or fault segments will be limited by neighboring ruptured VS segments, reducing seismic hazard.

Although our explanation links the first large induced earthquake on a fault distinctively to the long-term healing beforehand, fault healing actually strengthens the fault itself instead of promoting failure. The fault accumulates stress and elastic energy and becomes closer to failure only during human activities (Fig. 9a). If the same fault is being tectonically loaded at the same time, then the human activities would accelerate the (already existing) stress accumulation and eventually bring forward the nucleation of a later event. This type of "induced natural" or triggered earthquakes, as observed in some fluid injection sites in North America, instead show many similarities to natural interplate earthquakes such as stress drop[89]. In contrast to Groningen, these injection-induced earthquakes can be triggered by a pressure change of less than 0.1 MPa[71]. This suggests that the faults there are near critically stressed and healing might not be an important factor.

In summary, we exemplified that we can use the frictional properties and healing history to infer the seismic potential for induced earthquakes across the globe. This quantification shows that a better understanding of regional tectonic history is important when selecting sites for subsurface exploration across all reservoir settings. Notably, VS reservoir rocks healed for several to tens of million years, i.e., where faults have not been activated since Neogene times, may pose larger seismic hazards than currently anticipated. To select safer exploration sites, our results suggest to opt for VS rocks that heal slowly (e.g., $b < 0.005$) and for a relatively short period (Fig. 8). Still, it remains always best to avoid regions with large faults, because larger earthquakes might be triggered by human-induced (a)seismic events in the neighborhood. However, since VS faults will become non-seismogenic after one rupture, current tectonic settings or fault geometries that allow healed fault strength to be removed in smaller earthquakes could help reduce maximum earthquake magnitudes and corresponding earthquake hazard. For example, smaller earthquakes are promoted in settings with large geometrical and/or stress heterogeneities[90]. With properly selected operational strategies, the occurrence of smaller earthquakes can be

further enhanced[91]. In these settings, the occurrence of smaller earthquakes is actually beneficial, because ruptured fault segments will act as stronger barriers, limiting the size of subsequent events. Given that VS faults are still less hazardous than VW faults under the same conditions and thus often considered safer for sustainable exploration, Fig. 8 can be utilized to better estimate the maximum earthquake magnitude in advance. Yet, more accurately forecasting induced earthquakes, such as forecasting the duration of the aseismic period after human activities (Fig. 3), requires further integration of tectonic settings, material properties and operational conditions.

Beyond shear stress accumulation, human activities often pose more complexity and variability on seismicity than our model could predict, such as modifying the temporal and geographic statistics of seismicity[92]. Explanation of earthquake statistics would require a model of the complex fault network with fault roughness and heterogeneity taken into consideration, as well as the interaction and triggering between faults and fault segments. Towards induced earthquake forecasting, the model should also consider the dependence of healing on other factors such as shear load[93,94], healing behaviors deviating from Dieterich type such as power-law healing[95] or saturation of healing[96], as well as the dependence of weakening behavior on slip rates[97–100]. These mechanisms are not accounted for in the standard RSF framework. It would then be insightful to use microphysics-based models, which includes more physics to account for these processes[101–103].

To conclude, in contrast to current theory and operational strategies, earthquakes can nucleate on nominally stable VS fault segments as long as they have been able to heal at a reasonably high rate in an inactive tectonic environment over geological times. Over such time scales, faults have gained significant interface strength, expressed as increase in static frictions by around 0.25. Static strengths that are larger than currently used would delay anticipated fault failure, but allow for an adequate stress drop to obtain earthquake slip rates. Such more hazardous faults are described by a large RSF parameter $b$, which also leads to efficient (linear) frictional weakening with slip. Both RSF mechanisms facilitate the nucleation of induced earthquakes on VS faults and explain their similarities to induced earthquake nucleation on VW faults, in terms of nucleation stages and characteristic length scales. However, when considering seismic hazard, it is critical to account for a significantly reduced hazard for any subsequent event on the same fault segment. VS faults can no longer host earthquakes, because subsequent slip on human lifetimes is stable. VW fault segments may still nucleate earthquakes with sharply reduced stress drops, but the neighboring ruptured VS segments will impede rupture propagation and hence reduce anticipated future earthquake magnitudes. Our explanations and implications highlight the importance of a better understanding and quantification of healing in terms of its rate and duration. This is critical to correctly assess and communicate induced seismic hazards in all reservoir settings. Considering recent political and societal unrest, reliable hazard assessments may well be a key factor in deciding how much the shallow subsurface can contribute to transitioning toward a society driven by sustainable energy.

## Methods
### Model and solver
We developed and applied a numerical model that resolves dynamics on faults over both geological and earthquake time scales.

**Reservoir.** Earthquake sequences are simulated on a normal fault governed by RSF, crosscutting a depleting gas reservoir (Fig. 4). The setup and parameters follow preceding studies on the Zeerijp region in the Groningen gas field[30,104,105]. We restrict the research area to be between 2000 and 4000 m depth with a reservoir of 200 m thickness centered at 2950 m depth. The reservoir is cross-cut by a plane fault with a dip angle of 70 degrees. The fault offsets both sides of the

reservoir by 50 m. We cut the horizontal direction at 2000 m distance away from both sides of the fault plane. The setup is modeled in 2-D with the plane-strain assumption. The parameters are summarized in Supplementary Table 1.

The medium is assumed to be poroelastic. Under the quasi-dynamic approximation, we write out the momentum balance equations

$$\frac{\partial \sigma_{xx}}{\partial x} + \frac{\partial \sigma_{xz}}{\partial z} = 0$$
$$\frac{\partial \sigma_{xz}}{\partial x} + \frac{\partial \sigma_{zz}}{\partial z} = 0 \tag{3}$$

where subscripts $x$ and $z$ are the horizontal and vertical coordinates, respectively. $\sigma_{ij}(i, j = x, z)$ are the components of the total stress tensor denoting the total stress acting along the $j$ axis on the plane that is normal to the $i$ axis. Hooke's law relates the effective stress tensor to the displacement vector $(u_x, u_z)$ by

$$\sigma'_{xx} = (\lambda + 2G)\frac{\partial u_x}{\partial x} + \lambda \frac{\partial u_z}{\partial z}$$
$$\sigma'_{xz} = \lambda\left(\frac{\partial u_x}{\partial z} + \frac{\partial u_z}{\partial x}\right) \tag{4}$$
$$\sigma'_{zz} = \lambda\frac{\partial u_z}{\partial z} + (\lambda + 2G)\frac{\partial u_x}{\partial x}$$

with Lame's two constants $\lambda$ and $G$. The pore pressure $P$ connects the total stress and the effective stress via

$$\sigma'_{xx} = \sigma_{xx} - \alpha P$$
$$\sigma'_{xz} = \sigma_{xz} \tag{5}$$
$$\sigma'_{zz} = \sigma_{zz} - \alpha P$$

where $\alpha$ is the Biot coefficient. We prescribe constant stress boundary at the top of the model that satisfies the lithostatic pressure. The other three boundaries are left to be free-slip to mimic the far-away boundary.

For simplicity, we assume a homogeneous pressure change inside the reservoir due to gas extraction instead of modeling the corresponding fluid flow. No pore pressure change exists outside of the reservoir. During each timestep, the pressure change

$$\delta P = \dot{P}\delta t \tag{6}$$

where the pressure rate $\dot{P}$ is zero before gas production starts. It is assumed to be a constant that matches the average pressure change rate during the past sixty years of production in Groningen[50]. This simplification is justified based on the observation of relatively uniform depletion and little pressure difference across the faults[48]. In this way, we have applied a one-way coupling between pressure change and rock deformation, meaning that no feedback of rock deformation on pressure change is modeled, as gas is very compressible compared to the reservoir rock[30]. By assuming a homogeneous pressure change within the reservoir we ignore (i) the diffusion and convection taking place within the reservoir and (ii) the permeability of the fault interface and over- and under-burden. This approximation makes the instantaneous traction of the pressure front not possible. The interaction of the fluid pressure front and the rupture front is addressed in previous numerical and experimental works on induced seismicity[37,106]. Their methodologies and conclusions can be included in our models to obtain a further understanding of earthquake physics of this type.

The initial conditions are chosen according to the geological survey in Groningen (ref. 30 and reference therein). The stress condition follows the lithostatic pressure gradient with hydrostatic fluid and gas pressure. Gas is only present within the reservoir and this

results in an overpressure of 3 MPa, which agrees with the commonly observed value in Northern Netherlands[107].

**Fault.** The fault is assumed to be governed by RSF[7,8]. We use the regularized version to avoid near-zero singularity

$$\tau_s = a\sigma'_n \text{arcsinh}\left\{\frac{V}{2V_0}\exp\left[\frac{\mu_0}{a} + \frac{b}{a}\ln\left(\frac{\theta V_0}{D_{RS}}\right)\right]\right\} + \eta V. \tag{7}$$

The evolution of the state variable $\theta$ is governed by one among several different evolution descriptions, the aging law[8]

$$\dot{\theta} = 1 - \frac{V\theta}{D_{RS}}. \tag{8}$$

$\mu_0$ is the reference friction coefficient at the reference slip rate $V_0$, the characteristic slip distance is $D_{RS}$, and $a$ and $b$ are parameters that describe the relative influence of direct and evolutionary effects, respectively. The parameter $\eta = G/(2c_s)$ used in Equation (7) refers to the "radiation damping term" used in the quasi-dynamic approximation of inertia[108,109]. This term approximates the wave-radiated energy loss, which is significant when slip rate is higher than a seismic threshold

$$V_{seis} = 2\sigma'_n/\eta. \tag{9}$$

The fault functions as an additional interface condition on top of the boundary conditions mentioned above. Namely, the traction on either side of the fault needs to be equal and opposite so that the shear stress $\tau_s$ and effective normal stress $\sigma'_n$ can be solely defined on the fault plane as its projection. The slip rate $V$ is defined by the difference in tangential motion across the fault whereas difference in normal motion is fixed at zero as no fault opening is allowed. That is

$$\begin{aligned}
\tau_s &= \hat{n} \cdot \boldsymbol{\sigma}'|_{\Gamma^+} \cdot \hat{t} = \hat{n} \cdot \boldsymbol{\sigma}'|_{\Gamma^-} \cdot \hat{t} \\
\sigma_{n'} &= -\hat{n} \cdot \boldsymbol{\sigma}'|_{\Gamma^+} \cdot \hat{n} = -\hat{n} \cdot \boldsymbol{\sigma}'|_{\Gamma^-} \cdot \hat{n} \\
V &= \boldsymbol{u}|_{\Gamma^+} \cdot \hat{t} - \boldsymbol{u}|_{\Gamma^-} \cdot \hat{t} \\
0 &= \boldsymbol{u}|_{\Gamma^+} \cdot \hat{n} - \boldsymbol{u}|_{\Gamma^-} \cdot \hat{n}
\end{aligned} \tag{10}$$

where $\Gamma^+$ and $\Gamma^-$ refers to two sides of the fault interface, $\hat{n}$ and $\hat{t}$ are the unit vectors perpendicular and parallel to the fault, respectively. The second equation gets a negative sign because we use the convention that compressive normal stress is positive.

The initial stress on the fault matches the initial stress condition in the medium. The initial slip rate is set to be near zero due to the absence of tectonic activity before gas production. The state variable is calculated accordingly.

In Fig. 7 we introduced dynamic weakening due to flash heating into the model. This is realized by reformulating the RSF to incorporate a further reduction in frictional coefficient at high slip rates. We adopted the expression[110,111]

$$\tau_s = \frac{a\sigma'_n \text{arcsinh}\left\{\frac{V}{2V_0}\exp\left[\frac{\mu_0}{a} + \frac{b}{a}\ln\left(\frac{\theta V_0}{D_{RS}}\right)\right]\right\}}{1 + \frac{L}{\theta V_w}} + \eta V. \tag{11}$$

where $V_w$ is the velocity threshold when flash heating causes 50% reduction in steady-state friction coefficient. Namely, the effect of flash heating becomes notable (causes 1% reduction) at $0.01V_w$.

**0-D model.** We also use a simplified 0-D model, in which only the shallowest point on the fault inside the reservoir (marked in orange in Fig. 4) is modeled, for our parameter study. The effective normal stress and shear stress on the fault are a result of the poroelastic effect that is defined by the Biot coefficient, Poisson ratio and fault dipping angle

$$\begin{aligned}
\tau_s &= \tau_{s0} + \gamma_s P \\
\sigma'_n &= \sigma'_{n0} + \gamma_n P.
\end{aligned} \tag{12}$$

where $\tau_{s0}$ and $\sigma'_{n0}$ are the initial stresses, $\gamma_s$ and $\gamma_n$ are the two stress path parameters, which can be extracted from the loading curve in our 2D simulations[49,112]. This equation (12) and RSF equations (7) and (8) are the controlling equations of the 0-D model.

**Solver.** We use the MATLAB backslash direct solver for the PDEs stated above[113]. Spatial accuracy is satisfied by fully resolving the minimum nucleation length with ~ 20 grid elements[11,14]. Among the several nucleation length definitions[11,114], we have chosen the definition

$$\Lambda_0 = \frac{9\pi}{32}\frac{GD_{RS}}{b\sigma(1-\nu)} \tag{13}$$

because this definition is the smallest one under VW friction and is the only meaningful one under VS friction (Fig. 6). The size of $\Lambda_0$ is also called the cohesive zone length or the process zone length.

We use adaptive time stepping to handle the large variation of the slip velocity. Temporal accuracy is satisfied by restricting timestep size to be inversely proportional to slip rate[14]

$$\delta t = \zeta \frac{D_{RS}}{V_{max}} \tag{14}$$

where $V_{max}$ is the maximum slip rate on the fault and $\zeta$ is a factor controlled by the material and frictional parameters. The derivation of $\zeta$ follows ref. 14.

### Interface strength

For readers' convenience, we re-establish the definition of interface strength in ref. 47. The original definition was written with the help of the capital $\Theta$, here we rewrite it with the state variable $\theta$ for better readability. The relation between the two expressions is $\Theta = b\ln\frac{V_0 \theta}{D_{RS}}$.

The RSF defines the ratio between shear and normal stress $\tau/\sigma$ by slip velocity $V$ and the state variable $\theta$[7,8]

$$\mu = \tau_s/\sigma_n = \mu_0 + a\ln\left(\frac{V}{V_0}\right) + b\ln\left(\frac{\theta V_0}{D_{RS}}\right). \tag{15}$$

Rewriting RSF by solving for slip rate we obtain

$$\frac{V}{V_0} = \exp\left[\frac{\frac{\tau_s}{\sigma'_n} - \left(\mu_0 + b\ln\frac{V_0\theta}{D_{RS}}\right)}{a}\right]. \tag{16}$$

In this way, the slip rate is defined by the relative amplitude of the applied shear force $\frac{\tau_s}{\sigma'_n}$ compared to a state-dependent variable $\Psi = \mu_0 + b\ln\frac{V_0\theta}{D_{RS}}$. When the shear force is much higher than $\Psi$, a high slip rate is achieved, simulating the coseismic phase. If the shear force is much lower, a locked phase is simulated with a near-zero slip rate. Therefore the term $\Psi$ in the equation behaves just as well as a description of the fault strength and thus was termed as "interface strength"[47]. The first term of the aging law (Equation (8)) describes the healing behavior since the second term can be ignored when slip rate is near zero. In this case, we can integrate Equation (8) and substitute $\theta$ in the definition of interface strength (Equation (1)), which yields

$$\begin{aligned}
\theta &= \theta_i + t \\
\Psi &= \Psi_i + b\ln\left(1 + \frac{t}{\theta_i}\right)
\end{aligned} \tag{17}$$

where $\Psi_i$ and $\theta_i$ are the initial values of interface strength and state. The linear growth of the state variable and thus the logarithmic growth of the interface strength with time is therefore inherent in RSF.

Interface strength is only dependent on the state variable, which provides a memory of the fault loading and slip history via the aging law (Equation (8)). It is not to be confused with the steady-state friction

$$\mu_{ss} = \mu_0 + (a - b) \ln \frac{V}{V_0} \tag{18}$$

which is only meaningful when a steady state is achieved.

### Theoretical derivation of nucleation length $L_{III}$

We derive the theoretically predicted nucleation length $L_{III}$ in the scenario with a long healing time. This length scale is similar to what has been measured and defined as $2L_c$ in ref. 11. We follow the derivation there and demonstrate how the healing-induced fault strength increase makes nucleation under VS friction possible and how large the corresponding nucleation length is. The balance of fracture energy $G_c$ and consumed energy per crack growth $G_{mec}$ is used to estimate the nucleation length. For $0.5 < a/b < 1$, they predict that an aseismic growth to $2L_c$ is expected during nucleation (referred to as stage ii in this article). The fracture energy is estimated from the slip-weakening curve (Fig. 5c,f,i)

$$G_c = \frac{\sigma b D_{RS}}{2} \left( \ln \frac{V_f \theta_i}{D_{RS} \Omega_f} \right)^2, \tag{19}$$

where $V_f$ and $\theta_f$ are slip rate and state after the crack front arrival, respectively. $\Omega = V\theta/D_{RS}$ is a measure of how far the fault is from steady state. The consumed energy per crack length is estimated from the ambient-to-residual stress drop

$$G_{mec}(\mathcal{L}) = \frac{\pi \mathcal{L}}{2G'} \Delta \tau^2 \tag{20}$$

where $\mathcal{L}$ is the current crack length, $G' = G/(1 - \nu)$ is the equivalent shear modulus and $\Delta \tau$ is the ambient-to-residual stress drop, i.e. the difference between the ambient stress before crack arrival and the residual stress after stress drop. This was given as

$$\Delta \tau = -\sigma b \ln \Omega_f + \sigma(b - a) \ln \frac{V_f}{V_i} \tag{21}$$

where $V_i$ is the background slip rate which is controlled by the loading condition. This expression can only be positive if $a < b$, given $\Omega_f \sim 1$. This is reasonable and explains why natural earthquakes do not occur on VS faults. However, for induced seismicity, the reactivated fault did not experience a postseismic and interseismic phase and thus the initial stress is higher than the steady-state ambient stress used in ref. 11. Taken this into consideration, the deviation from steady-state $\Omega_i = V_i \theta_i / L$ before nucleation should be added to Equation (21), therefore

$$\Delta \tau = \sigma b \ln \Omega_i + Eq.(21) = \sigma b \ln \frac{\Omega_i}{\Omega_f} - \sigma(a - b) \ln \frac{V_f}{V_i}. \tag{22}$$

Since $\theta_i$ is large after a long healing time, this expression can become positive even when $a > b$. This explains from another perspective why induced earthquakes are possible on VS faults. By equating $G_c$ and $G_{mec}$, we have

$$L_{III} = 2L_c = \frac{2}{\pi} L_b \left[ \frac{\ln \Omega_i/\Omega_f + \ln V_f/V_i}{\ln \Omega_i/\Omega_f - \frac{a-b}{b} \ln V_f/V_i} \right]^2 \tag{23}$$

which is the length of aseismic nucleation growth.

For a better understanding of this expression, we look at several limit cases. We assume $\ln \Omega_i/\Omega_f$ and $\ln V_f/V_i$ are positive for simplicity. If $\ln \Omega_i/\Omega_f << \ln V_f/V_i$ and $a < b$, this is exactly the assumption[11] made. The limiting value of $L_{III}$ is $L_\infty = \frac{2}{\pi} \left( \frac{b}{b-a} \right)^2 L_b$. Otherwise, we notice that the value in the bracket in (23) is always larger than 1 and increases as $a/b$ increases. This indicates that, similar as ref. 11, an aseismic growth of nucleation length to $L_{III}$ should also be expected when $a > b$. Even when $0.5 < a/b < 1$, the not-to-be-ignored $\Omega_i$ of induced seismicity makes the value of $L_{III}$ different than ref. 11 suggested.

Due to the limited spatial extension of the reservoir width in our study, the nucleation might not be able to achieve its maximum length as $L_{III}$ predicts. If this is the case, the maximum nucleation length will be the reservoir width (Fig. 5). This incomplete nucleation often results in a lower maximum slip rate. Additional simulations are conducted to show the dependence of $L_{III}$ on frictional parameters. For example, the nucleation length increases when $b$ is decreased, as shown in Fig. 4.

## Data availability

The associated simulation data files for this research can be accessed at: https://doi.org/10.5281/zenodo.8172688.

## Code availability

The associated code for this research can be accessed at: https://bitbucket.org/lmuu/indnuc/src/master/.

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

## Acknowledgements

This study is part of the "InFocus: An Integrated Approach to Estimating Fault Slip Occurrence" project (grant number: DEEP.NL.2018.037, M.L, A.R.N, Y.v.D) funded by NWO's (Dutch research council) DeepNL program. We thank Daniel Faulkner, Jean-Philippe Avouac, Jean-Paul Ampuero, Yuntao Ji, Chris Spiers, Yajing Liu, Yuji Itoh, Binhao Wang, Liviu Matenco and Femke Vossepoel for discussions and suggestions.

## Author contributions

M.L., A.R.N. and Y.v.D. conceptualized the project, developed the methodology, analyzed the results and reviewed the manuscript. M.L. developed the software, visualized the results and drafted the original manuscript. A.R.N. and Y.v.D. supervised the project.

## Competing interests

The authors declare no competing interests.
