## [Transparent Peer Review file · Nature Communications]

Why induced earthquakes occur on conventionally stable faults

Corresponding Author: Dr Meng Li

Version 0:

Reviewer comments:

Reviewer #1

(Remarks to the Author)

I appreciate the authors for responding the comments in the revised version. The revised manuscript has significantly improved and has clarified most of the unclear parts. I have no further comments and the paper can be accepted for publication.

(Remarks on code availability)

Reviewer #2

(Remarks to the Author)

Review of Meng Li et al "Why induced earthquakes occur on conventionally stable faults: frictional healing explains"

This is my (re-)review of the revised manuscript submitted to Nature Communications. I previously reviewed a version submitted to Nature Geoscience. I continue to hold the opinion that this interesting manuscript explores an important observation – that velocity strengthening or frictionally stable faults (whose frictional strength increases with slip speed) are able to rupture seismically. The authors demonstrate through a suite of 0D and 2D numerical models that even velocity strengthening faults can nucleate 'one-off' ruptures if they're allowed to heal for a sufficiently long duration, on the order of thousands to millions of years. After the one-and-done seismic event, the velocity strengthening faults in their models subsequently creep aseismically in the absence of external perturbations. This idea deserves recognition in an impactful journal like Nature Communications. The authors have addressed most of my previous concerns in this revision. I have a few further comments on topics I feel strongly about (approximately in order of decreasing importance), and hope that addressing these will help improve clarity and broaden the reach of this manuscript.

1. In section 3.3 (Nature of RSF parameter b) or elsewhere – It is probably important to spell out here that the RSF ' b ' parameter is being used as a proxy for healing rate, because this may not be immediately obvious to a reader that is not thinking about RSF regularly, especially as this proxy relationship is central to the conclusions of the paper. However, this brings up an interesting point that I had not considered in my previous review. Some studies (e.g., Nakatani and Mochizuki, 1996, GRL; Karner and Marone, 1998, GRL; Ryan et al., 2018, JGR: Solid Earth; Dillavou and Rubinstein, 2020, PRL) show that healing is dependent on shear load, with significantly higher healing occurring at low shear loads (or "true stationary contacts"). This is not immediately transferrable to RSF or to the modeling being done in this study, because it probably implies that the RSF ' b ' and frictional state depend on the absolute value of the tectonic load experienced by the fault in some complex way. I am curious if the authors can speculate on the effect of this "shear stress dependent healing" on their results, and incorporate some discussions surrounding this.

2. Lines 92 – 93: "...even for planar faults devoid of external perturbations." By phrasing this sentence as they have, the authors reveal an important assumption and potentially a limitation of this study – that the models represent isolated planar faults, whereas natural faults are usually accompanied by complex damage zones and fault networks, even in induced seismicity settings. It would be nice to see this acknowledged at the beginning or end of the manuscript.

3. Lines 109 - 110: "Our model produced aseismic slip events after laboratory timescales of healing". I am not sure that I am

seeing this in any of the figures. Also, how is aseismic vs. seismic slip discriminated here?

4. Lines 476 – 479: This is only true for earthquakes in inter-plate regions. There are intra-plate settings (e.g., continental normal faults) which have such low loading rates (0.01-0.1 mm/yr) and long recurrence times (thousands of years) that the proposed mechanisms of long healing times may also be relevant there.

(Remarks on code availability)

Reviewer #3

(Remarks to the Author)

Dear Editor, I have reviewed “Why induced earthquakes occur on conventionally stable faults: frictional healing explains” by Li and coauthors.

The manuscript is well written and illustrates nice models, based on the well-known Rate-State Friction (RSF) framework. The authors model a fault undergoing a geologically fast perturbation after a prolonged period of healing and characterized by velocity strengthening friction simulating the case of a reservoir depletion. Beside some interesting, but quite specialist discussion on RSF parameters and how they compare with other measurements of cohesion, the key point of the manuscript is the illustration of how fault healing may be the mechanism to trigger (induced) earthquakes on velocity-strengthening faults.

The approach of the Authors is rigorous, and their analysis well worth the attention of the specialists. There is comparatively much less literature on modelling healing with the RSF framework, than modelling fault slip velocity changes. It is also worth a the observation that slow healing rates may prevent multiple unstable reactivations, but in turn favour one (or few!) shocks upon reactivation, favouring subsequent stable slip. This is a nice perspective for the reader, which turns on the spotlights on mismatch between the timeframe of human exploitation of reservoirs and the geological times needed to heal some faults (not only those long inactive though!).

I have two lines of considerations about the present manuscript. One about the “tone” of the manuscript and one (quite minor) about the actual science and points of discussion.

First of all, I am not particularly surprised by the notion that fault healing is what allows unstable fault slip nucleation and more generally allows for the seismic cycle to happen. It looks like that also the previous reviewers were on the same line. A considerable amount of theoretical and experimental work is present on the topic, some of it already cited in the manuscript (Marone et al., 1998 and Beeler 2004 provide nice summaries, but many other exist, as the Authors well know). Therefore claiming this as a conceptual novelty, starting from the title, is sort of awkward to read.

For instance claiming things like “Even though healing is inherent in rate-and-state friction (RSF), it has not been only been connected to seismicity for the first time last year, where an ultralow healing rate was used to explain the slow slip events in the Hikurangi subduction zone [45]” imply that the analysis of healing with RSF is somehow neglected. I would tend to regard both ref. 45 (Shreedaran et al., 2023), in which RSF healing is applied to slow slips, and the present manuscript, in which you apply similar concepts to induced seismicity, as specific applications of the already general RSF model. Not a novel approach or concept to the matter of earthquake nucleation.

Also much of the premises for the claimed novelty are based on the “common knowledge” that velocity-strengthening faults are considered “stable” and unable to nucleate earthquakes.

Although this might be considered a valid point by a non-specialized geological/geophysical audience, it is fundamentally wrong and quite clear to people working with RSF in the laboratory or with numerical modelling.

Nucleation can happen even on velocity-strengthening faults either spontaneously, in case of a strong rheological contrast and/or slip localization (e.g. Volpe et al., 2024 Science Advances) or because the fault in its initial state is quite far from the “steady” slipping state to which the conventional RSF analysis applies, as also outlined by the previous reviewers.

Making a RSF-based model of unstable slip in a velocity-strengthening fault is quite straightforward, and it is not much different from making a nucleation via a “normal” slip weakening. And in fact, your “interface strength” parameter that degrades with “b” is substantially equivalent to a cohesion degrading with slip. Yes your model adds some nice details on the different nucleation stages in velocity-strengthening vs. velocity weakening faults and there are interesting discussions, but in definitive I fail to grasp how modelling healing and fault slip/slip rate with RSF is conceptually novel.

Finally, the authors claim to clarify the physical mechanisms of healing and earthquake nucleation, but I would argue that making models with RSF hardly elucidates any physics at all. After all even RSF parameters are just parameters measured in the lab of which we still poorly understand the underlying physical mechanism.

As I said, I like the manuscript, which is rigorous and interesting but, given the above considerations, I find it should aimed at the specialist reader, rather than a generalist audience.

From the scientific point of view, I have only very few observations, as the Authors have thought revised the manuscript already. The application of the model to reservoirs and to induced seismicity is very interesting but I would suggest to put some “caveats”. The present manuscript puts an emphasis on the seismic risk, saying that fault reactivation of velocity-strengthening yet slowly healing faults may trigger only once. However if the faults are velocity neutral, this may induce a “few” events (fig. 2) . these “few” events that may occur on human timescales and might be quite damaging quite important. Also, from a human perspective perspective, the choice of those b values is critical to assess the risk. The Authors know well how variable RSF parameters, can be variable in the laboratory, being experts in the field. I would suggest to be more cautious with the “one-shot” faults, but suggest to the reader that individual faults may become increasingly stable after the initial reactivation.

It should be noted also that the choice of RSF parameter values in your model, albeit based on laboratory experiments, ignore the option that other healing mechanism may occur in natural faults. This may render relatively low healing (but still

more healed than what laboratory rates suggest) vs. high healing faults basically indistinguishable from a human perspective, while assessing the risk. I would add this nuance to the manuscript discussion on healing.

Another point worth of discussion is that your analysis considers only single fault patches (based on mechanical stratigraphy) but and neglects the general structural and mechanical heterogeneity that large faults have (e.g. Faulkner et al., 2003; Tesei et al., 2014; Bullock et al., 2014; Marchesini et al., 2022; Arts et al., 2024). Also, minor faults abutting on the main fault may be difficult to recognize at the reservoir scale, both in geophysical images and even in drill cores. Minor faults may have less-developed and less phyllosilicate-rich fault cores (hence less velocity strengthening) but nonetheless may have the potential for seismic slip or the necessary stored energy to provide the stress drop useful to trigger the main fault. In this scenario, the RSF model is too simplistic. This does not imply that the model is not worth developing, but a discussion of the possible real-world complications is needed.

Figure S4 NEEDS to be integrated in the main text to make it clearer and easier to read.

Line comments

14: I think the word “paradox” should be changed. As explained by other reviewers and by the authors themselves, it is fairly understood that velocity strengthening friction applies at steady conditions (during the nucleation) but does not prevents unstable fault slip in absolute terms).

19-20: I would be more cautious here. I suggest something like “subsequent slip on human lifetimes is likely to become increasingly quiet/stable”

23-26: If you have the space in the abstract, you could specify HOW the site choice is affected by the insights provided by your work.

35: I would argue that the load is high but the strain rate is low, as it should be in plate interiors that “transmit” stress from the plate boundaries.

70-71: I think this statement is broadly incorrect. VS may hinder slip nucleation, but may not prevent it (as also demonstrated by the other reviewer). For instance, when the nucleation patch is grown enough, the increasing frictional resistance behind the crack tip may not be enough to stop the nucleation. I would try to re-formulate this statement with a bit more nuance.

105: Also grain indentation by pressure-solution may contribute to cohesion and compaction.

104-106: the sentence is awkward to read. Maybe something like “cohesive rocks show evidence of cementation at the microscale”?

171-121: this is a rather strange statement. Healing is commonly known as the cause for the accumulation of new elastic energy for the seismic cycle. First is not entirely new that slow slip/creep has been linked to low healing only last year (e.g. Carpenter et al., 2011, Nature Geoscience; Tesei et al., 2012; JGR; and others, including some work from the Utrecht group!). Second, healing is inherent in RSF, as you said. Therefore, even if it was not explicitly applied to the specific case of induced seismicity, the RSF formulation should hold for all general cases, if you believe is a solid framework for fault friction modeling.

317: Sometimes negative b is measured in the lab and, usually it is coupled with negative healing rates (albeit in short timescale-experiments). the possibility of negative b and effectively zero healing could be briefly considered with a couple of lines here.

355: it is unclear here what “large enough” means here. Are you referring to a magnitude threshold, a minimum moment release, minimum slip, a simple earthquake that can brak the surface...etc. Please specify.

373-375: I would keep the original caveats (now deleted) that the analysis relies on the knowledge of frictional properties of the fault materials and that the last time of activity can be confidently estimated.

405: 66 is not the correct reference for pointing the reader to carbonate-hosted earthquakes. Provide a couple of other citations.

3.3

This whole paragraph about the nature of the “ b ” parameter is a bit speculative and does not add much to the paper. I suggest it can be safely deleted.

450: this is not surprising at all, since “ b ” is a fit parameter to describe the evolution of stress with slip or velocity. if it's positive, it describes some form of slip weakening.

460: also Pozzi et al., 2023 Geophysical Journal International <https://doi.org/10.1093/gji/ggad322>

510-512: This statement is rather sharp. I suggest to rephrase it in a more dubitative way. either because faults contain heterogeneous material (both VS and VW/neutral), there are minor faults, and because your analysis has limitations that might not always make it applicable to nature.

(Remarks on code availability)

Version 1:

Reviewer comments:

Reviewer #2

(Remarks to the Author)

The authors have now broadly addressed all of my concerns and I have no further revisions to request before publication.

I only have one suggestion, and I leave it to the editor and authors to accept or ignore - I think the current title of this study is very vague, contains no results, and has a somewhat "clickbait" nature to it - it signals to the reader that they must dig deeper to even begin to understand the essence of what this manuscript is about. I would encourage the authors to distill into their title the key result from this work. For example "Numerical insights into the role of healing in induced seismicity on conventionally stable faults" or similar immediately informs the reader that this is a chiefly numerical study that explores frictional healing. To be clear, I'm not specifically suggesting that the authors adopt my example title, but that they could think about a more representative title for this neat study.

(Remarks on code availability)

Reviewer #3

(Remarks to the Author)

Dear Authors,

I appreciate the revised version of the manuscript. It has significantly improved, with respect to the version I initially read. It is a solid and very interesting work.

I am still not totally convinced about the conceptual novelty, but in my opinion it is publication-ready.

(Remarks on code availability)

Utrecht, July 2, 2024

Concerns: Submission of revised manuscript “Why induced earthquakes occur on conventionally stable faults: frictional healing explains”

Dear Editor and reviewers,

Please find enclosed our revised manuscript “**Why induced earthquakes occur on conventionally stable faults: frictional healing explains**”. This is a re-submission of our original manuscript submitted to *Nature Geoscience* (manuscript number NGS-2023-09-01852-T) following constructive reviews. We adequately addressed all comments and extended and clarified explanations. The major changes are summarized below:

- A) We highlighted the novelty of our work with previous works better acknowledged. We are the first to use healing to explain induced seismicity. Although the mechanisms that activate velocity-strengthening faults are known, whether damaging earthquakes (i.e., fast slips) will happen has not been studied, not to mention that some studies even lack the ability to separate earthquakes and aseismic slips after fault activation. Previous numerical and experimental studies often stop at acknowledging transient (a-)seismic slips due to a substantial amount of initial perturbation (*Dieterich, JGR Solid Earth, 1979; Perfettini and Ampuero, JGR Solid Earth, 2008*), while they failed to propose a proper mechanism that allows such perturbation and a quantified overview of the nucleation process.
- B) We provided a complete picture of earthquake nucleation on velocity-strengthening faults. We improved the description of the nucleation process in the revision and proposed new nucleation length scales for earthquakes after healing. These length scales are good proxies that allow easier comparison and application of our model to other scenarios and settings. Thus, our theoretical expressions of the length scales can be applied and verified in future numerical and observational studies.
- C) We extended the discussions to highlight the applicability of our hypothesis from velocity-strengthening faults to conventionally unstable velocity-weakening faults, and from induced to natural earthquakes. We improved our narration from experimental and observational perspectives with quantitative comparisons in order to reach a broader audience.

We thank the reviewers for their constructive and useful comments, which we all carefully and considerately addressed. Below we provide a detailed point-by-point response to all points. The revised manuscript with all changes tracked is attached for clarity. We hope this thoroughly revised manuscript can be reconsidered for publication.

Sincerely yours,

Meng Li (corresponding author), on behalf of the co-authors

Response to reviewer 1:

Induced Seismicity has been a major challenge for many industrial activities including the hydrocarbon recovery, geothermal energy developments and underground storage. Fluid injection or extraction can awaken dormant faults and trigger earthquakes. The submitted manuscript provides a new explanation to the earthquake nucleation on velocity strengthening faults, which are not favorable for seismic slip. The rate and state friction theory has been applied to reproduce the behavior of velocity strengthening of faults in the laboratory and to extrapolate it to the real time scales. The paper is well-structured and well-written. The proposed methodology is clear and the results are well-justified. However, I am not convinced if the findings are a significant breakthrough and are suitable for publishing in Nature Geoscience. The following are my most important concerns:

We thank the reviewer for the affirmative comment on our paper. Below clarify the novelty of our work and the physics we propose, because they appear to not be well understood by the reviewer.

1. Fault healing is not a new finding and many authors have presented it in previous publication. The application of this concept to induced seismicity is the novelty of this work, which might not be sufficient for a paper in Nature Geoscience.

Our novelty indeed lies in using fault healing to explain the seismogenesis of induced seismicity. Fault healing has been observed in experimental studies and described with rate-and-state friction since Dieterich (1979). Despite the long history of experimental studies, its application to the real world is not given enough attention, but that is now starting to change. We think this novel application to a broadly studied field is sufficient for publication in Nature Geoscience or Communications. Recently fault healing is for the first time applied to explain slow slip events observed in the Hikurangi subduction zone, which has been published in **Science** last year (Shreedharan et al., Science, 2023). Induced seismicity arguably encompasses a larger field than slow slip events, and here we explain a long and critical paradox that is of critical importance for society and the energy transition that is on many peoples minds. Besides justifying the impact of healing on induced seismic hazards, we also provide a quantitative description of the nucleation on healed fault for the first time, which is crucial and even decisive on the subsequent physical processes.

Dieterich J H. Modeling of rock friction: 1. Experimental results and constitutive equations[J]. Journal of Geophysical Research: Solid Earth, 1979, 84(B5): 2161-2168.

Shreedharan S, Saffer D, Wallace L M, et al. Ultralow frictional healing explains recurring slow slip events[J]. Science, 2023, 379(6633): 712-717.

2. Aseismic slip during fluid injection or reservoir creep during fluid extraction can trigger seismic slip (e.g. Guglielmi et al. 2015). Aseismic deformation in Groningen has also

previously studied by van Wees et al. 2017. Aseismic slip occurs on velocity strengthening faults that can result in seismic failure. *This* can be also a potential explanation for why the earthquakes occur on Velocity Strengthening VS faults. So, the physics behind the occurrence of induced earthquakes on VS faults is already known.

Guglielmi, Y., Cappa, F., Avouac, J.-P., Henry, P. & Elsworth, D. Seismicity triggered by fluid injection-induced aseismic slip. *Science* 348, 1224–1226 (2015).

van Wees, J.-D., Osinga, S., Van Thienen-Visser, K. & Fokker, P. A. Reservoir creep and induced seismicity: inferences from geomechanical modeling of gas depletion in the Groningen field. *Geophys. J. Int.* 212, 1487–1497 (2017).

We disagree, because the fundamentals of earthquake mechanics are not correctly used. The fact that aseismic slip can occur on velocity-strengthening (VS) faults does not mean that seismic slip – i.e., earthquakes – can occur as well. This is supported by a large body of earthquake physics work (e.g., Ziv, 2007; Perfettini & Ampuero, 2008). Moreover, the presence of aseismic and seismic slips on the same fault is not an explanation or the physical mechanism behind an earthquake, either. This is why a paradox exists, which is ignored in virtually all other works – also in Groningen. In fact, a large number of numerical models used simplified frictional formulation or did not use any frictional formulation to simplify or force an artificial earthquake nucleation. Those works could not generate or separate seismic slips. However, when planning subsurface activities we need to know if and when earthquakes, instead of aseismic slips will occur. This is our motivation to explain an mechanism of induced seismicity with a particular attention on its spontaneous nucleation. On top of the physical mechanism, we also provide a quantitative study on the impact of healing time and healing rate on the subsequent induced slip rates.

Moreover, the cited references are not really applicable.

- The Guglielmi 2015 paper discusses several possibilities to generate aseismic and/or seismic slips with fluid injection. Particularly, their experiment intended to model a heavily injected scenario, with low effective normal stress. Mechanisms discussed there, e.g., dilatancy and fault opening, are not causative processes occurring in fluid-depletion reservoirs, where effective normal stress is increased instead.
- van Wees et al. (2017/8) does not study aseismic deformation in Groningen. Their only reference to “aseismic” is: “*Part of the slip may occur aseismically. In this simplified approach, the dynamic effects of slip and slip weakening are discarded, and it is assumed that all incremental slip is released seismically.*” It is clearly stated that they did not model the dynamics after fault failure. In general, papers of van Wees and co-authors and many others lack the ability to model self-consistent nucleation and thus distinguish whether an earthquake (i.e., seismic slip) or an aseismic event will occur.

This illustrates that the physics behind the occurrence of induced earthquakes on VS faults is not known and still needs to be clarified. We updated the introduction with a better acknowledgement to previous studies to better explain the fundamentals and the novelty (line 70-91).

Perfettini H, Ampuero J P. Dynamics of a velocity strengthening fault region: Implications for slow earthquakes and postseismic slip[J]. Journal of Geophysical Research: Solid Earth, 2008, 113(B9).

Ziv A. On the nucleation of creep and the interaction between creep and seismic slip on rate-and state-dependent faults[J]. Geophysical research letters, 2007, 34(15).

3. The authors suggest that the occurrence of induced earthquakes can prevent subsequent earthquakes, since the ruptured portion forms a barrier that inhibits future larger and more damaging earthquakes. Is it possible to define a magnitude threshold, beyond which the fault healing would prevent the occurrence of subsequent earthquakes? In this case, if a M1 earthquake occurs on a fault, one must not anticipate a M5? Is it reasonable?

No. It is currently not possible to do so. Our statement on that no subsequent earthquake will occur on the same velocity-strengthening (VS) fault is based on the observation that VS faults are not relocked after the first seismic event. In this way, the ruptured fault portion will not be able to nucleate subsequent earthquake nor can it be ruptured by neighboring earthquakes. However, we believe the question being asked by the reviewer must be answered in the perspective of a fault network, instead of a single fault. A direct answer could be: yes, if an M1 earthquake occurs on a fault, that same fault would have a very low chance to nucleate an M5 earthquake afterwards.

Yet, an M5 earthquake could still occur on other faults in the network. And the unruptured fault portion, if any, could still be ruptured later due to ruptures jumping over. Certainly, the larger the ruptured portion is in the first event, the more difficult it will be to jump over it. In our group we are currently working on a Linear Elastic Fracture Mechanics formulation building forth on e.g.. Weng and Ampuero, JGR, 2019, to efficiently model a whole fault network. The question can be better answered there because that is beyond the scope of this work. However, the current unquantified concept is already fully novel and critical in order to assess future seismicity in induced seismicity settings.

Weng H, Ampuero J P. The dynamics of elongated earthquake ruptures[J]. Journal of Geophysical Research: Solid Earth, 2019, 124(8): 8584-8610.

4. The main question in the context of induced seismicity is the ability to forecast, mitigate or control the risk of triggering large-magnitude events. It is not clear how these findings can contribute to the current efforts in the energy transition?

We answer this question from two perspectives. First, we show that earthquakes, instead of aseismic events, can occur on VS faults, while people currently think they can not. This is a critical difference, which needs to reach a large audience. The regions with VS faults, which are targeted for their safety in sustainable explorations, are more dangerous than currently

anticipated. Moreover, healing during a long geological period of time also increases seismic potential of VW faults. Hence regions with a long inactive tectonic history and a high healing rate is generally more dangerous than thought. Our explanation highlights the importance of a better understanding and quantification of fault healing rate and healing time. This is critical in assessing induced seismic hazards in all reservoir settings.

Second, we did present a quantitative study on the impact of healing and frictional properties on the seismic slip rate. We also discussed the seismic potential of several representative reservoir lithology. This is our first step towards hazard assessment as earthquake physicists. Without a better hazard assessment, we cannot guarantee a safe transition towards sustainable energy. For sure a comprehensive hazard or risk estimate should have considered more ingredients such as sediment damping. However, we believe we have provided an important building block for moving towards mitigating induced seismic hazards. We have clarified the importance and impact in the updated manuscript (abstract, results 2.3, discussions 3.1, 3.4 and summary).

Response to reviewer 2:

This interesting manuscript explores an intriguing problem – the observation that velocity strengthening or frictionally stable faults (whose frictional strength increases with slip speed) are able to rupture seismically. The authors demonstrate through a suite of 0D and 2D numerical models that even velocity strengthening faults can nucleate ‘one-off’ ruptures if they’re allowed to heal for a sufficiently long duration, on the order of thousands to millions of years. After the one-and-done seismic event, the velocity strengthening faults in their models subsequently creep aseismically in the absence of external perturbations. This intriguing proposition put forward in this manuscript certainly deserves recognition in an impactful journal such as Nature Geosciences. However, I have a few questions, comments, and suggestions which should be addressed before this well-written manuscript is ready for publication. I hope my comments (approximately in order of decreasing importance) help improve clarity and broaden the reach of this manuscript.

We thank the reviewer for the positive comments on the impact and importance of our paper.

1. The overarching hypothesis in this manuscript is that velocity strengthening faults can nucleate single earthquakes after healing over thousands or millions of years, and then will never nucleate an instability again (unless they’re perturbed or their frictional properties change significantly etc.). This is presented as a surprising finding, but I’m not so sure that I’m sufficiently surprised by this.

I say this because in the context of rate-and-state friction (RSF), velocity strengthening (VS) faults cannot spontaneously nucleate an instability in response to infinitesimal perturbations, but RSF imposes no restriction on the ability of a VS fault to nucleate an instability when it is very far from steady-state. In the case of healing over thousands of years, the fault has been pushed far from steady-state, i.e., θ (in Fig. 2D for example) $\gg D_c/V_0$, because there is limited far-field tectonic loading to be overcome. In this case, I think there is no expectation of steady or stable creep once the shear stresses on the fault overcome its strength even on VS faults...and this should be better articulated in the manuscript.

We fully agree that faults cannot remain stable when human activities are to perturb them. We should have emphasized more that our work aimed to answer whether an earthquake or an aseismic event (in the form of e.g., slow slip events) would occur afterwards. This depends on the healing time and healing rate (proportional to RSF parameter b). We have explicitly spelt this out in our abstract, conclusion and main text. To support the statement, we updated Fig. 2 by adding another set of simulation on a slowly healing fault for comparison. The slowly healing fault is activated to slip aseismically despite having the same “ θ ” variable as the rapidly healing fault.

2. The overall manuscript is written in a way that implies that this process of ‘one-off’ earthquakes on a VS fault is uniquely applicable for induced seismicity but is this true? In other words, do we really need millions of years of healing to produce such a rupture on a VS fault? To me, the proposed framework is extremely similar to conventional laboratory slide-hold-slide experiments that are done to infer healing (e.g. refs 37 and 61 in the manuscript).

So, I went ahead and simulated a 0D spring-block slider SHS model with the same VS friction as in the red case of Fig. 2, and after just a 1000 s hold, I see an instability followed by damped oscillations (similar to your green curve).

This implies that it is not very difficult to generate an instability on a VS fault within the RSF framework as long as the fault is pushed sufficiently far from steady-state, and my intuition is that this could happen quite routinely in other tectonic settings as well as long as there is a mechanism to push the fault far from steady state after every instability.

Here below are the answered to each question asked.

- (1) Our hypothesis is indeed based on and extending the findings of SHS experiments. As your spring-slider simulation suggested, it is not difficult to generate aseismic “damped oscillations” on VS faults if a mechanism to push the fault far from steady state exists. This is also communicated in earlier works e.g., Ziv, 2007; Perfettini & Ampuero, 2008. However, it is unknown whether and under what condition the instability on VS faults would grow into earthquakes i.e., seismic slips, as well as the natural origins of such perturbation. The damped oscillations can thus be taken as the counterpart to the aseismic events in nature, such as slow slip events, in case of low healing rate or short healing time. In fact, recently an ultralow healing rate has indeed been linked to the slow slip events observed in the Hikurangi subduction zone (Shreedharan et al., Science, 2023). The novelty of our work is that we found that for faults with high healing rates,

seismic slips are possible after a geological timescale of healing. As mentioned in the answer above, we have made this point clear by adding another set of simulations on a slowly healing fault to Fig. 2 for comparison (section 2.1).

- (2) Another distinct difference between our work and the 0-D spring-slider model is the spontaneous nucleation and the related length scales we modeled in 2-D. We extended the results and added a new Fig. 4 for a better description (section 2.2).
- (3) The mechanism proposed is indeed not uniquely applicable to induced earthquake, but may also happen for natural earthquakes. However, the role of healing is decisive for intra-plate faults, where tectonic loading rate is slow, to gain enough strength that can be converted to large stress drops and fast slips later. Natural earthquakes are recurring and have relatively short recurrence intervals. Hence we think fast coseismic and postseismic healing mechanisms should be more decisive in tectonically active regions (Heaton et al., 1990; Bedford et al., 2023). Interseismic Dieterich-type healing, with logarithmic growth with time, could well be negligible in terms of their contribution to strength recovery. This, in combination with other differences between induced and natural earthquakes, is better communicated in a new discussion subsection 3.4 with a separate new Figure 6 added.
- (4) We agree with your intuition that this kind of (a-)seismic events could happen quite routinely in other tectonic settings, as we have replied in point (1) and (3) above.

Perfettini H, Ampuero J P. Dynamics of a velocity strengthening fault region: Implications for slow earthquakes and postseismic slip[J]. *Journal of Geophysical Research: Solid Earth*, 2008, 113(B9).

Ziv A. On the nucleation of creep and the interaction between creep and seismic slip on rate-and state-dependent faults[J]. *Geophysical research letters*, 2007, 34(15).

Heaton T H. Evidence for and implications of self-healing pulses of slip in earthquake rupture[J]. *Physics of the Earth and Planetary Interiors*, 1990, 64(1): 1-20.

Bedford J D, Hirose T, Hamada Y. Rapid fault healing after seismic slip[J]. *Journal of Geophysical Research: Solid Earth*, 2023, 128(6): e2023JB026706.

Shreedharan S, Saffer D, Wallace L M, et al. Ultralow frictional healing explains recurring slow slip events[J]. *Science*, 2023, 379(6633): 712-717.

3. Figure 2E and associated discussions in-text are quite interesting to me. There is a log-linear relationship between max slip rate and duration of static hold with “no production”. This result is, of course, very similar to the SHS. However, I couldn’t find this connection clarified or further elaborated on in-text and I think it is a missed opportunity to strengthen an obvious connection between lab expts and numerical models.

Together with adding another set of simulations on a slowly healing fault, we updated the visualization of Fig. 2E, 2F. We find a log-linear growth of maximum slip rate with healing time for the earthquakes on the rapidly healing fault, while the growth of max slip rate for

the aseismic events on the slowly healing fault is linear with healing time. However, a logarithmic growth of stress drop with healing time is observed on both types of faults. This logarithmic growth of stress drop is what we usually observe in experimental studies, but accurate slip rate measurements are not always possible, especially for fast events. We added the comparison with laboratory data as suggested in the text with the support of Fig S1. We briefly added the reason of the different patterns of the slip rate growth (line 163-172): the inertial effect (the loss of wave-radiated energy), which is prominent in the fast events, could suppress the slip rates (Thomas et al., 2014).

Thomas M Y, Lapusta N, Noda H, et al. Quasi-dynamic versus fully dynamic simulations of earthquakes and aseismic slip with and without enhanced coseismic weakening[J]. Journal of Geophysical Research: Solid Earth, 2014, 119(3): 1986-2004.

4. Numerous recent studies have demonstrated that the frictional stability parameter (a-b) often tends to be velocity dependent (Ikari and Saffer, G-Cubed, 2011; Rabinowitz et al., G-Cubed, 2018; Shreedharan et al., Im et al., Nat. Geo., 2020; G-Cubed, 2022 just to name a few that I remember now). If we consider a VS fault with velocity dependent (a-b), it will make the fault more VS at the onset of an instability and quickly quench it. I think this is an important consideration and caveat in the hypothesis presented here and should be appropriately acknowledged.

We agree. On top of that, experimental studies also revealed more complicated healing behaviors. It would be insightful to consider how healing affects subsequent earthquake nucleation using microphysics-based models. These thoughts are briefly added to the discussions due to word limit (section 3.3).

The remaining comments are minor, and in order of their appearance in the manuscript:

5. Line 46 – 48: These lines refer to nucleation, propagation, and arrest whereas the subsequent lines abruptly transition to an articulation of RSF. It is important to mention here that RSF usually only describes nucleation, and to a degree, propagation, i.e., range of velocities that are pre-onset of dynamic weakening.

We added the typical loading velocity range in velocity-step experiments: from 1 $\mu\text{m/s}$ to 1 mm/s (line 56).

6. Line 55: Classical stability analysis indicates that instabilities only nucleate under VW friction...this is only partially correct as classical stability analysis only refers to spontaneously arising instabilities in response to tiny perturbations from steady-state.

Here our description is only restricted to what classical stability analysis indicates. By definition, stability analysis refers to the stability of an object in response to tiny (or

infinitesimal) perturbations. As for large perturbations, we added two sentences in the end of the next paragraph (line 83-91).

7. Line 66: This paragraph starts with “However,” but continues to validate the previous paragraph rather than setting up a contradiction, so I’m unsure that “however” is necessary/correct here.

We deleted the word “however”.

8. Line 106: This is the first mention of “pressure”, and therefore should clarify that this refers to pore pressure.

We replaced the word “pressure” by “pore pressure”.

9. Lines 117-118: “..several slow-slip events still occur.” I assume the authors are referring to the damped oscillations in green, and if so, this should be stated. Also, an aside, what is the criteria for classifying a rupture as slow vs fast here?

Yes, we added “(VN, a=b)” at the end of this sentence to refer to the velocity-neutral fault. We added a bracket “(cm/s to m/s)” after referring to “seismic rate” here. The accurate definition comes later in section 2.2 to facilitate the statement there (line 211).

10. Line 121: “and thus behaves as a barrier for future seismicity” Based on the text/models presented here, it can only be inferred that the VS fault will not nucleate a future event as long as the fault remains at steady-state. Whether it will serve as a “barrier” to future seismicity or allow a future rupture to propagate through it will also depend on its dynamic friction and fracture energy, does it not?

We have rephrased as suggested, also in the abstract and conclusion.

11. Lines 128-129: “Specifically, maximum slip rate grows logarithmically with the applied healing time” This statement states the opposite of what is in Fig. 2E and I assume this is a typo. The maximum slip rate increases linearly for logarithmic increase in the applied healing time or “no production” time.

Fig. 2E shows that the maximum slip rate increases linearly for exponential increase in the applied healing time. Hence our original formulation is an equivalent expression.

12. Lines 202-204: Again, see my comments 1 and 6.

Answered above.

13. Line 234: Is it necessary for $b > 0.01$ for the VS rocks to generate EQs or is this simply the range selected for simulations. If b has to be greater than 0.01, I'm curious as to why this is the case, and its physical implications.

This is the range of b under the simulated set of parameters (elastic modulus, D_{RS} , etc). It is here to declare that a high healing rate is required to generate earthquakes in this kind of rock after a geological timescale of healing. If b is too low, the required healing time will be unreasonably longer. After careful investigation, we updated the range to ($b > 0.005$).

14. Lines 289-291: "Since no subsequent..... more damaging earthquakes". The way this is written implies that the geologic volume is somehow safer after an induced earthquake because the fault can only creep aseismically in the future. This seems to be a strong claim to make based on data presented here. The dynamic rupture could fundamentally alter the frictional properties of the fault surface, it could heal faster thus pushing it far from steady-state etc. This should be rephrased to incorporate the idea that this is contingent upon the fault not being pushed far from steady-state again.

We rephrased as answered above in #10, with the emphasis that healing in human timescale is not enough to make the fault seismogenic again. The discussions on other possible healing mechanisms, as well as other possibilities to push the fault far from stable again, are included in the extended discussion section (sections 3.2, 3.3).

Response to reviewer 3:

This paper offers an intriguing examination of the persistent question regarding the occurrence of induced earthquakes along faults, juxtaposed against the velocity-strengthening frictional properties observed in laboratory experiments. Although the authors present their ideas as novel, it is worth mentioning that similar concepts have been previously explored, notably by Rice and Gu in 1983, who demonstrated that perturbing a seismic source at its zero-load point can result in behavior akin to seismic slip, even with velocity-strengthening friction. Despite this, the paper contributes meaningfully to the dialogue surrounding this critical issue. However, the manuscript posits a theory that, as far as I can tell, could predict a highly variable stress drop across between induced and tectonic events. This is a premise that may not fully align with existing observational data. I believe this needs more discussion, examination, and clarification within the paper (see my major comments)

Rice, J.R., Gu, Jc. Earthquake aftereffects and triggered seismic phenomena. *PAGEOPH* 121, 187–219 (1983). <https://doi.org/10.1007/BF02590135>

We thank the reviewer for mentioning the previous work on velocity-strengthening (VS) friction. We have added the paper of Rice & Gu 1983 in the introduction. Our work is based on and extending these works to demonstrate that (1) earthquakes, instead of aseismic slips, can nucleate on VS fault segments, as long as they have been able to heal in an inactive tectonic environment on a historical to geological timescale; (2) this effect of healing that increases seismic potential is critical for induced seismicity, no matter fault friction is VW or VS; (3) nucleation process exhibits several critical length scales, some of them are also applicable to VS faults. These are our novel points.

We should clarify that our work does not imply a highly variable stress drop nor a difference in stress drop between induced and natural earthquakes. We have shown that stress drop of an induced *event* can be variable due to different healing time and healing rate. However, depending on the size of the stress drop, in combination with other parameters, that induced event can achieve fast slips or remain aseismic. Therefore the stress drop of induced earthquakes (i.e., fast slips) does not need to be highly variable. With this clarified, it also becomes clear that we do not predict a difference in stress drop between induced and tectonic events, either. Despite that, we thank the reviewer for pointing out the importance of using “stress drop” to connect our work with observational studies. We have strengthened the connection in discussions. We respond with a detailed answer below at question #2.

Major comments:

1. The discussion around lines 128-130 delves into the crux of the paper, focusing on the

proposed mechanism. This segment, pivotal to the paper's argument, would greatly benefit from a clearer articulation of the hypothesis in terms that are accessible to a broader audience. Consider incorporating a simplified explanation of the core hypothesis into the abstract to enhance comprehensibility. In my opinion, the current emphasis on 'healing time' may obfuscate the fundamental argument. It appears that the essence of the ability to induce seismic events in a velocity-strengthening fault lies in the increased stress drop associated with greater healing. Stress drop, being a parameter that is, to some extent, observable, should be more prominent in the paper.

We adopt the idea to use “stress drop” as a proxy to explain the effect of healing. We have updated the abstract and conclusion to make it more prominent. Fig. 2F and the related description are updated with a comparison between rapidly and slowly healing faults to emphasize this concept. The discussion on stress drop is also extended and connected to observational studies for a better reach to a broader audience (section 3.2).

2. Interpreting Figure 4 suggests a significant influence of healing time on the stress drop associated with an event, which, if aligned with the paper's theory, implies a distinction between the stress drop characteristics of intraplate induced seismicity and that of tectonic seismicity. However, Reference 43 appears to challenge this implication, not supporting a marked difference in stress drop between these seismicity types. Additionally, Figure 2B indicates a roughly twofold difference in stress drop between velocity-strengthening and weakening conditions for identical healing durations. It is crucial for the paper to elucidate the expected variations in stress drop as per the proposed theory and to critically evaluate the plausibility of these variations in light of observational evidence. This will not only clarify the theoretical underpinnings but also facilitate a better understanding of the theory's applicability and validity from an observational standpoint.

We thank the reviewer for pointing out the potential confusions in comprehending Fig. 4. We continue our reply to the main comment here. As said, Fig. 4A (new Fig 5) exhibits varying stress drop in induced events caused by varied healing time and healing rate. But not all the events are earthquakes, some are aseismic events without fast slips. We could imagine that the stress drops of those earthquakes are not much variable, because sufficient stress drop is needed to achieve fast slips (Fig. 2E, F). Actually this is evident by the earthquakes shown in Fig. 3 in a few VW or VS scenarios, which do not show much difference in terms of stress drop.

Fig. 2B, as pointed out, does show a roughly twofold difference in stress drop between VW and VS fault friction in the simplified 0-D model. But even that is well expected in our opinion. Earthquakes observed in natural and induced setups generally have a stress drop ranging between 0.1 and 10 MPa. Similar to Ref. 43 induced earthquakes in Groningen are also exhibiting similar stress drops as natural earthquakes. Stress drops from 0.1 MPa to up to several MPa are often reported (a summary of several observational studies can be

found in Ameri et al., 2020). A two-fold difference in stress drop is thus very normal. We now included the similarity of our simulated stress drop to these studies. We also compared the gained strength due to healing in our model to the experimental and observational studies on cohesion in the discussions 3.2 to facilitate the understanding from an observational perspective.

The second point to be made is that our work does not imply a difference in stress drop between induced and natural earthquakes. Fig. 4 (new Fig. 5) is only about induced events. It is not suitable to be applied to tectonic setups to question if a short healing time there would result in a markedly lower stress drop. Different from induced earthquakes, tectonic earthquakes are recurring. Hence fast coseismic and postseismic healing mechanisms should be more decisive in building fault strength in tectonically active regions (Heaton et al., 1990; Bedford et al., 2023). We extended this comparison between induced and tectonic seismicity in the discussions 3.4 with a separate new Fig. 6.

Ameri G, Martin C, Oth A. Ground-motion attenuation, stress drop, and directivity of induced events in the Groningen gas field by spectral inversion of borehole records[J]. *Bulletin of the Seismological Society of America*, 2020, 110(5): 2077-2094.

Heaton T H. Evidence for and implications of self-healing pulses of slip in earthquake rupture[J]. *Physics of the Earth and Planetary Interiors*, 1990, 64(1): 1-20.

Bedford J D, Hirose T, Hamada Y. Rapid fault healing after seismic slip[J]. *Journal of Geophysical Research: Solid Earth*, 2023, 128(6): e2023JB026706.

Minor comments:

3. The assertion in lines 55-58 is not quite true; the statement is more accurately limited to small perturbations around a steady state. Even in that case, systems can be destabilized at steady state by introducing additional physics, such as asymmetries in geometry or material contrasts, which could be significant in contexts like Groningen. It would be beneficial for the authors to explore and reference literature on these aspects to provide a more nuanced perspective.

Aldam M, Bar-Sinai Y, Svetlizky I, Brener EA, Fineberg J, Bouchbinder E. Frictional sliding without geometrical reflection symmetry. *Physical Review X*. 2016 Oct 28;6(4):041023.

Rice JR, Lapusta N, Ranjith K. Rate and state dependent friction and the stability of sliding between elastically deformable solids. *Journal of the Mechanics and Physics of Solids*. 2001 Sep 1;49(9):1865-98.

(see section 4)

Stability analysis, by its definition, only refers to the stability of an object in response to tiny (or infinitesimal) perturbations. Hence we find the assertion there suitable. Instead of expanding the description here, we have added two sentences in the end of the next paragraph to discuss large perturbations (line 83-91).

We thank the reviewer to point out the uncovered mechanisms such as fault geometry and material contrast. The objective of our work is to show that even with the simplest setup, healing could still contribute, and even play a principal role, to make VS faults seismogenic. Additional physics such as geometry is significant to consider, but beyond the scope of this work. Therefore we decide not to expand our narration in this introductory paragraph to distract the readers, but to add a concise summarizing sentence to acknowledge related works (line 87-89).

4. Lines 148-152: The language used in this section appears overly casual and simplified, particularly the phrase "human activities just speed up...". It is important to convey that human interventions might not merely accelerate existing processes but could also alter the system's behavior fundamentally. The term "planned natural events" is also not suitable for describing seismic occurrences, as it implies a level of predictability and intention that is not present in natural seismic events. A more accurate representation would acknowledge the complexity and variability of seismic events, potentially influenced by human activities in ways that extend beyond simple acceleration. This includes changes to seismic properties, such as the b-value, rather than merely advancing the timing of events. Moreover, the statement in the context of North America, may not be valid, where many instances of induced seismicity are observed in intraplate regions characterized by low natural stressing rates.

Agreed. We replaced "a planned natural event" by "a later natural event" to avoid the nuance of predictability. We rephrased the north America statement to avoid indicating it as the only explanation. This part of discussion is extracted from the results and reformulated into a separate discussed section 3.4 (line 416-429).

5. In Figure 2, panel E, there appears to be a discrepancy in terminology that could lead to confusion among readers. The panel discusses the initiation of production, yet the caption refers to this as "healing time." Given that panels A-E denote the start of production at time 0. Ensuring consistency in the terms used to describe the start of production and "healing time" will enhance clarity and prevent potential misunderstandings.

We believe that the legend has made such confusion. Actually the horizontal axis of Fig. 2E is shared with 2F to show the time past since the last fault activity, i.e., the healing time. We have updated the visualization and removed the confusing legend.

6. Regarding the reservoir model described around line 585, the term "poroelastic" might not be the most accurate descriptor, given that the model does not incorporate key aspects of poroelasticity, such as diffusion and two-way coupling. Describing the model as elastic and accounting for undrained conditions seems more appropriate. However, this approach is likely reasonable for the model's intended application.

Thanks for the confirmation of the model validity. We adopted the term “poroelastic” because the model accounted for the deformation (mainly compaction) due to pore pressure change, despite using one-way coupling and homogeneous pressure depletion. Since compaction is the major poroelastic effect, we will keep using this term.

7. The mathematical representation, specifically equations 4 and 5, there appears to be an inconsistency in the usage of ΔP and P . It is important to clarify whether there is a distinct difference between these terms as used in the equations. If they are intended to represent different concepts (e.g., a change in pressure versus absolute pressure), this distinction should be explicitly stated to avoid confusion.

We use the notation of P to refer to absolute pore pressure and ΔP the pore pressure change (Fig. 2). ΔP used in Eq. 5 in the Methods section is indeed confusing because it is used to explain how we modify fault pressure numerically in a single timestep Δt . To avoid the confusion, we now use δP to refer to the the pressure change in a single timestep δt in Eq. 5.

Utrecht, March 12, 2025

Concerns: Submission of revised manuscript “Why induced earthquakes occur on conventionally stable faults”

Dear Editor and reviewers,

Please find enclosed our revised manuscript “**Why induced earthquakes occur on conventionally stable faults**”. This is a re-submission of our original manuscript (manuscript number NCOMMS-24-40224-T) following constructive reviews. We addressed all comments and extended and clarified explanations. The major changes are summarized below:

- (1) We reconstructed and improved the results and discussion sections to make the manuscript better readable for a non-expert and generalist audience. We strengthened the link and broadened the implications of our work to natural observations. We also focused on refining the tone in response to some reviewer comments.
- (2) We emphasized the knowledge gap in the introduction and clarified the methodological and scientific novelty of our work throughout the manuscript and particularly in the abstract and conclusion. In summary, our novelties are:

For general readers from academia and beyond:

- a. We simulate and explain that single non-recurring earthquakes can nucleate on velocity-strengthening faults, due to healing that increases fault strength continuously over millions of years. This is facilitated by the introduction of a numerical method that for the first time simulates a fault continuously from millions of years (healing) down to milliseconds (earthquakes).
- b. We show an increased hazard of the first induced earthquake and reduced hazards of subsequent events, on both velocity-weakening and -strengthening faults. This feature could help forecast and mitigate induced seismic risk.

For geologists and geophysicists:

- c. We show that the failure of the first induced earthquake occurs at a fault strength significantly higher (+0.25) than currently used static friction. This indicates that induced earthquakes would happen on stronger and less critical faults than currently anticipated.
- d. We reveal how nucleation occurs on velocity-strengthening faults. Induced earthquakes are similar to regular earthquakes on velocity-weakening faults but have unique nucleation stages and length scales.

We thank the reviewers for their constructive and useful comments. We think that our attempt to carefully and considerately address them improved the manuscript significantly. Below we provide a detailed point-by-point response to all points. The revised manuscript with all changes tracked is attached and the line numbers there are used in the responses. We hope this thoroughly revised manuscript can be reconsidered for publication.

Sincerely yours,
Meng Li (corresponding author), on behalf of the co-authors

Response to Reviewer #1:

I appreciate the authors for responding the comments in the revised version. The revised manuscript has significantly improved and has clarified most of the unclear parts. I have no further comments and the paper can be accepted for publication.

We thank the Reviewer for the endorsement. This round we have made improvements in the discussions and small edits elsewhere based on other reviewers' comments, while leaving the reviewed results and conclusions as robust and concrete as before.

Response to Reviewer #2:

This is my (re-)review of the revised manuscript submitted to Nature Communications. I previously reviewed a version submitted to Nature Geoscience. I continue to hold the opinion that this interesting manuscript explores an important observation – that velocity strengthening or frictionally stable faults (whose frictional strength increases with slip speed) are able to rupture seismically. The authors demonstrate through a suite of 0D and 2D numerical models that even velocity strengthening faults can nucleate ‘one-off’ ruptures if they’re allowed to heal for a sufficiently long duration, on the order of thousands to millions of years. After the one-and-done seismic event, the velocity strengthening faults in their models subsequently creep aseismically in the absence of external perturbations. This idea deserves recognition in an impactful journal like Nature Communications. The authors have addressed most of my previous concerns in this revision. I have a few further comments on topics I feel strongly about (approximately in order of decreasing importance), and hope that addressing these will help improve clarity and broaden the reach of this manuscript.

We thank the Reviewer for confirming that the importance and novelty of our idea justifies publication in Nature Communications.

1. In section 3.3 (Nature of RSF parameter b) or elsewhere – It is probably important to spell out here that the RSF ‘ b ’ parameter is being used as a proxy for healing rate, because this may not be immediately obvious to a reader that is not thinking about RSF regularly, especially as this proxy relationship is central to the conclusions of the paper. However, this brings up an interesting point that I had not considered in my previous review. Some studies (e.g., Nakatani and Mochizuki, 1996, GRL; Karner and Marone, 1998, GRL; Ryan et al., 2018, JGR: Solid Earth; Dillavou and Rubinstein, 2020, PRL) show that healing is dependent on shear load, with significantly higher healing occurring at low shear loads (or “true stationary contacts”). This is not immediately transferrable to RSF or to the modeling being done in this study, because it probably implies that the RSF ‘ b ’ and frictional state depend on the absolute value of the tectonic load experienced by the fault in some complex way. I am curious if the authors can speculate on the effect of this “shear stress dependent healing” on their results, and incorporate some discussions surrounding this.

We have clearly spelt out the function of RSF b as a proxy for healing rate twice in result section 2.1. First as measured and suggested in laboratory experiment (line 162-163 in the tracked-change manuscript, same below), and second time after the theoretical definition of “interface strength” using equations (line 314).

We agree that our simplified model needs to be examined in a broader context: for example, by a more thorough investigation of the link between the b value derived from velocity steps experiments and the one obtained from slide-hold-slide experiments.

However, this is beyond the scope of the current manuscript. In response to Reviewer #3, we deleted this section 3.3, which includes mainly specialist-oriented discussions, to highlight our messages to the general audience. Therefore, we mentioned the shear stress dependence and other factors influencing healing (line 772-776) but didn't include further speculation.

2. Lines 92 – 93: "...even for planar faults devoid of external perturbations." By phrasing this sentence as they have, the authors reveal an important assumption and potentially a limitation of this study – that the models represent isolated planar faults, whereas natural faults are usually accompanied by complex damage zones and fault networks, even in induced seismicity settings. It would be nice to see this acknowledged at the beginning or end of the manuscript.

We rephrased the paragraph to demonstrate that this assumption is not a limitation of the study, but a novelty instead. Previous studies have attempted to attribute the instability on VS faults to external factors while frictional healing, which is inherent in every fault, was ignored. We use a simple fault setup to highlight the importance of healing here.

We agree it is important to connect our findings to the real world. We have newly added the implication of our model on the induced seismicity in a complex fault network (line 718-725) and indicated the importance of the presence of fault network on other observations such as earthquake statistics (line 766-771).

3. Lines 109 - 110: "Our model produced aseismic slip events after laboratory timescales of healing". I am not sure that I am seeing this in any of the figures. Also, how is aseismic vs. seismic slip discriminated here?

This was shown in a supplementary figure. We now moved that supplementary figure showing the comparison with experiments to the main text, which comes in as the new Figure 2. We added a note that seismic and aseismic slips are discriminated by a common slip rate threshold at that location (line 167). A quantified expression of the seismic threshold ($V_{seis} = a\sigma_n/\eta$, Rubin and Amupero, 2005) is defined in the next section after the frictional parameters and nucleation stages have been introduced (line 346).

4. Lines 476 – 479: This is only true for earthquakes in inter-plate regions. There are intra-plate settings (e.g., continental normal faults) which have such low loading rates (0.01-0.1 mm/yr) and long recurrence times (thousands of years) that the proposed mechanisms of long healing times may also be relevant there.

We limited our comparisons to inter-plate natural earthquakes and added "interplate" to section titles and figures. The mechanism of intraplate earthquakes is uncertain and likely involve other mechanisms beyond the scope of this study.

Response to Reviewer #3:

The manuscript is well written and illustrates nice models, based on the well-known Rate-State Friction (RSF) framework. The authors model a fault undergoing a geologically fast perturbation after a prolonged period of healing and characterized by velocity strengthening friction simulating the case of a reservoir depletion. Beside some interesting, but quite specialist discussion on RSF parameters and how they compare with other measurements of cohesion, the key point of the manuscript is the illustration of how fault healing may be the mechanism to trigger (induced) earthquakes on velocity-strengthening faults.

The approach of the Authors is rigorous, and their analysis well worth the attention of the specialists. There is comparatively much less literature on modelling healing with the RSF framework, than modelling fault slip velocity changes. It is also worth a the observation that slow healing rates may prevent multiple unstable reactivations, but in turn favour one (or few!) shocks upon reactivation, favouring subsequent stable slip. This is a nice perspective for the reader, which turns on the spotlights on mismatch between the timeframe of human exploitation of reservoirs and the geological times needed to heal some faults (not only those long inactive though!).

We thank the Reviewer for highlighting one of the novelties of our work that healing has been less focused or frequently ignored and its impact. We highlighted it in our renovated problem statement paragraph in the Introduction (line 118-142 in the tracked-change manuscript, same below).

We thank the Reviewer for pointing out the implication of our results on seismic hazard, especially the subsequent hazard after the first event. We extend our discussions on this perspective and link it to the difference between induced and natural earthquakes, so that the mismatch between the short time scale of human activities and the long healing period becomes lucid. We updated Figure 9.

Besides making these stronger connections to the real world, we condensed the discussions for specialists so that the manuscript is better oriented to general audience.

I have two lines of considerations about the present manuscript. One about the “tone” of the manuscript and one (quite minor) about the actual science and points of discussion.

We have updated the manuscript thoroughly and refined the “tone”. See below our point-to-point response.

First of all, I am not particularly surprised by the notion that fault healing is what allows unstable fault slip nucleation and more generally allows for the seismic cycle to happen. It looks like that also the previous reviewers were on the same line. A considerable amount of

theoretical and experimental work is present on the topic, some of it already cited in the manuscript (Marone et al., 1998 and Beeler 2004 provide nice summaries, but many other exist, as the Authors well know). Therefore claiming this as a conceptual novelty, starting from the title, is sort of awkward to read.

Healing has never been linked or used to explain induced earthquakes. Healing has indeed been investigated, but our conceptual novelty lies in the explanation and quantification of mechanisms and the occurrence of the one-off induced earthquakes: why, when and in how many stages do earthquakes nucleate on velocity-strengthening (VS) faults. This is critical and an unknown result that both specialists and professionals in the field need to know to better evaluate induced seismic hazard.

For example, the Reviewer commented: “healing allows for the seismic cycle to happen”. From our results, it becomes clear that this is not correct. In induced seismicity settings not that many earthquakes could follow in cycle, but rather it is the occurrence of the first (one-off) earthquakes on each fault segment that are more hazardous. This has never been demonstrated in any study. The other two reviewers agree on this. This is a message that needs to penetrate to the general public and particularly professionals working on the energy transition and on hazard from induced seismicity. We also need to note here that healing is not the cause of induced earthquakes. We replied to a comment below (page 15) to clarify.

Nevertheless, we agree that the title was not accurate. We removed the second half of the title “frictional healing explains”. The research question was twofold: why velocity-strengthening faults are unstable, and why and how instabilities on such faults evolve into earthquakes. Our novel quantification answers both and shows that the mechanism is the combination of the long-term healing and the subsequent slip-weakening friction. Both are inherent in RSF for faults with a large ‘b’ parameter. We also updated the abstract with a more accurate statement.

We respectfully disagree that our novelty has been claimed by previous works and the other reviewers agree with that. We added a paragraph in the Introduction dedicated to better demonstrating the knowledge gap and the novelty (line 118-142):

- (1) The concept of stable VS faults is still the standard for the very large majority of specialists and non-specialists despite a few pioneering numerical and experimental works. That might be because the mechanisms and consequences are not understood. We provide that understanding and demonstrate that these mechanisms can be widely present across the globe, particularly in induced seismicity settings.
- (2) Previous works do not answer why and if instabilities could grow into earthquakes. Actually, several works only simulated aseismic pulses without fast seismic slips, such as Volpe et al, 2024, Science, as mentioned by the Reviewer later (page 10).
- (3) Previous works do not identify and justify the source of their proposed “external perturbation” to trigger instabilities on VS faults. We demonstrate that such perturbation is not necessary.

The always present fault healing is enough to trigger earthquakes, not only aseismic events. Since it is always present it has implications for many places around the world.

- (4) Previous work did not identify the large discrepancy in stress drop and earthquake magnitude between the first induced earthquake and subsequent events. This difference has large implications for estimating the maximum magnitude and seismic hazard.
- (5) We established quantitative descriptions of induced earthquake nucleation and rupture behaviors on VS faults in numerical and theoretical manners.

We tackle these points and provide the understanding and explanations required for both non-specialists (1-4) and specialists (5) to accept this important possibility for induced earthquakes. To ensure that this possibility is accounted for in hazard assessment by non-specialists, and can thereby help the energy transition, we think it is crucial to publish this work in a journal well read by non-specialists. We also optimized the abstract for better understanding by a broad, non-expert audience.

For instance claiming things like “Even though healing is inherent in rate-and-state friction (RSF), it has not been only been connected to seismicity for the first time last year, where an ultralow healing rate was used to explain the slow slip events in the Hikurangi subduction zone [45]” imply that the analysis of healing with RSF is somehow neglected. I would tend to regard both ref. 45 (Shreedaran et al., 2023), in which RSF healing is applied to slow slips, and the present manuscript, in which you apply similar concepts to induced seismicity, as specific applications of the already general RSF model. Not a novel approach or concept to the matter of earthquake nucleation.

The application of healing to understand induced earthquakes in this work as well as the slow slips as presented by Shreedaran et al., 2023 are not just exercises of known knowledge, as the Reviewer commented from the beginning: “there is comparatively much less literature on modelling healing with the RSF framework”. Healing has been ignored for a long time in earthquake models. However, there is a reason behind why integrating healing to the mechanism of nucleation is not part of the standard approach for natural earthquakes and shouldn't have been. In discussions, we explained that healing can be ignored in natural interplate earthquakes due to the short recurrence interval and really only takes effect in scenarios such as intraplate induced seismicity (this work) or slow slip events under low tectonic loading (Shreedaran et al., 2023, Volpe et al., 2024) (line 702-709). Only now that induced seismicity is becoming a significant societal concern and unequivocal laboratory data are present to demonstrate the VS friction on specific faults, it becomes critical to really account for this.

Also much of the premises for the claimed novelty are based on the “common knowledge” that velocity-strengthening faults are considered “stable” and unable to nucleate earthquakes.

Although this might be considered a valid point by a non-specialized geological/geophysical audience, it is fundamentally wrong and quite clear to people working with RSF in the laboratory or with numerical modelling.

We totally agree that the common knowledge of stable VS faults is wrong. Unfortunately, as pointed out by the Reviewer, it is still perceived or assumed as such by nonspecialists and even most specialists. We think this justifies why our work is suitable for Nature Communications. We as three authors, two specialists in numerical modeling and one in laboratory experiments, all having been working with RSF for many years, were still surprised by some of our findings. Moreover, over the last 3-4 years we have discussed these findings on meetings with many leading scientists, who consistently have been very interested in our solution to the induced earthquakes observed on stable faults. Examples include Jean-Philippe Avouac, Jean-Paul Ampuero and Daniel Faulkner (conferences and during the first author's doctoral defense, where most questions were about this chapter) and scientists such as Jan-Dirk Jansen who have used classic Mohr-Coulomb failure for induced earthquakes (emails and conversations). It was also frequently perceived that one-off earthquakes were caused by a sudden and large enough perturbation and/or because the initial condition was pushed far from steady state. We have shown that both conditions are unnecessary and incorrect (see also our response to the next comment). This is why we urgently need to circulate our work to both the general public and the specialists.

Nucleation can happen even on velocity-strengthening faults either spontaneously, in case of a strong rheological contrast and/or slip localization (e.g. Volpe et al., 2024 Science Advances) or because the fault in its initial state is quite far from the "steady" slipping state to which the conventional RSF analysis applies, as also outlined by the previous reviewers. Making a RSF-based model of unstable slip in a velocity-strengthening fault is quite straightforward, and it is not much different from making a nucleation via a "normal" slip weakening. And in fact, your "interface strength" parameter that degrades with "b" is substantially equivalent to a cohesion degrading with slip. Yes your model adds some nice details on the different nucleation stages in velocity-strengthening vs. velocity weakening faults and there are interesting discussions, but in definitive I fail to grasp how modelling healing and fault slip/slip rate with RSF is conceptually novel.

We thank the Reviewer for the confirmation on our contribution to quantitatively understanding the fault dynamics through defining the nucleation stages and length scales.

The Reviewer suggested that VS faults can become unstable after perturbations caused by rheological contrast or slip localization and proposed the work of Volpe et al., 2024. But instability is only the first step towards seismicity. It does not explain why such instability would grow into earthquakes instead of staying aseismic, which is the missing piece of the puzzle necessary to explain induced seismic ruptures. Even the stick-slip events in the

small-scale experiments in Volpe et al., 2024 did not become seismic. We clarified this key novelty in our research question statement (line 120-123), as we have replied earlier on page 7.

Moreover, using different mechanisms to explain induced seismicity would lead to completely different predictions that could be verified later. If “a strong rheological contrast”, as proposed by the Reviewer, could be the major factor, one would expect repeating earthquakes on the same fault since such contrast does not disappear with time, whereas our mechanism would predict a one-off event since human timescale is not enough for the fault to re-heal, after the healed strength has been removed by the first event. This finding that after the nucleation of one earthquake, subsequent earthquakes cannot occur is fully novel and of critical information for professionals working in subsurface exploration as well as society as a whole. Therefore, this work is not “adding nice details”.

The Reviewer also suggested that VS faults can become unstable if the initial condition is far from steady state of RSF. We have rebutted this point in the previous comment. In fact, our models started from steady state. Moreover, it doesn't matter where the models start from. The departure of the simulated faults from steady state happens during human activities, not during healing.

The Reviewer suggested a similarity between the proposed “interface strength” and cohesion. In fact, we have compared our strength gain due to healing and the cohesion values measured in laboratory. We indeed find our interface strength gain comparable to the measured intact rock cohesion, probably because the strength of the healed fault in our study is close to that of an intact rock. However, this does not indicate that frictional healing is the same thing as cohesion, as they have different physical mechanisms, which is at least reflected from their different dependency on normal stress. To avoid confusion, we removed the comparison to cohesion in the text.

Finally, the authors claim to clarify the physical mechanisms of healing and earthquake nucleation, but I would argue that making models with RSF hardly elucidates any physics at all. After all even RSF parameters are just parameters measured in the lab of which we still poorly understand the underlying physical mechanism.

We claim that we clarified the physical mechanism of induced earthquake nucleation. But we did not clarify the physical mechanism of healing. We are sorry we have made such impression to the Reviewer. We restricted our simulations to the RSF framework and elaborated how such simple friction formulation can already explain the induced earthquakes in Groningen and many other places around the world successfully.

Despite RSF being a phenomenological law, we and our colleagues never stopped searching for the underlying physics. We have microphysics-based models (e.g. Chen-

Niemeijer-Spiers (CNS) model) to support and supplement RSF. RSF has demonstrated its advantage over other friction formulations, say slip-weakening, in describing the physics of earthquake nucleation. We agree that the limitation of RSF (and the healing description inherent in RSF) and the implication must be specified. We have rewritten the last paragraph of discussion dedicated to this before the concluding remarks (line 772-778).

As I said, I like the manuscript, which is rigorous and interesting but, given the above considerations, I find it should aimed at the specialist reader, rather than a generalist audience.

We disagree. The Reviewer has agreed in a previous comment (on page 8) that correcting the “common knowledge” on stable VS faults is why this work targets a more general audience. Moreover, the reviewer rather argued that specialist readers already know a part of these findings. If that was true, why would this paper then need to aim at them again?

Based on the responses given above and by the positive comments of the other two reviewers about the novelty and general applicability of our work, we think that the conclusions in this work should reach both the general audience and the specialists because the criticality of these findings for subsurface usage for the energy transition is driven solely by non-experts. Our work should reach a general audience, in order to facilitate progress towards using the subsurface to transition our entire society to sustainable energy. Induced seismic hazard and societal response thereto are arguably the largest inhibitor of using the subsurface. Our findings are so relevant and fundamental that we cannot think of other novel findings that would justify a broad dispersion of induced seismicity results.

To make these connections crystal clear, we strengthened the link of our results to the real world and specified our suggestions to the site selection of subsurface exploitations (line 740-765).

From the scientific point of view, I have only very few observations, as the Authors have thoroughlt revised the manuscript already. The application of the model to reservoirs and to induced seismicity is very interesting but I would suggest to put some “caveats”. The present manuscript puts an emphasis on the seismic risk, saying that fault reactivation of velocity-strengthening yet slowly healing faults may trigger only once. However if the faults are velocity neutral, this may induce a “few” events (fig. 2) . these “few” events that may occur on human timescales and might be quite damaging quite important. Also, from a human perspective perspective, the choice of those b values is critical to assess the risk. The Authors know well how variable RSF parameters, can be variable in the laboratory, being experts in the field. I would suggest to be more cautious with the “one-shot” faults, but suggest to the reader that individual faults may become increasingly stable after the initial reactivation.

We agree the subsequent hazard following the first earthquake is equally important to discuss. We added such discussion on the damaging earthquake sequence on VW faults (line 695-705) and addressed the earthquake triggering between faults and fault segments if VS and VW faults coexist (line 718-725). To maintain accuracy, we have clarified in the abstract and the text that the "one-shots" faults only occur on VS faults, not on velocity-neutral (VN) faults. On VN faults, slow slip events still follow, but not reaching seismic rates in our simulations (Fig. 3). We have described this observation for VN faults (line 243-244). Given that VN ($a=b$) is such a thin critical state we find no reason to cloud our conclusion by mixing VN and VS faults.

It should be noted also that the choice of RSF parameter values in your model, albeit based on laboratory experiments, ignore the option that other healing mechanism may occur in natural faults. This may render relatively low healing (but still more healed than what laboratory rates suggest) vs. high healing faults basically indistinguishable from a human perspective, while assessing the risk. I would add this nuance to the manuscript discussion on healing.

We acknowledged these healing behaviors (line 772-778).

Another point worth of discussion is that your analysis considers only single fault patches (based on mechanical stratigraphy) but and neglects the general structural and mechanical heterogeneity that large faults have (e.g. Faulkner et al., 2003; Tesei et al., 2014; Bullock et al., 2014; Marchesini et al., 2022; Arts et al., 2024). Also, minor faults abutting on the main fault may be difficult to recognize at the reservoir scale, both in geophysical images and even in drill cores. Minor faults may have less-developed and less phyllosilicate-rich fault cores (hence less velocity strengthening) but nonetheless may have the potential for seismic slip or the necessary stored energy to provide the stress drop useful to trigger the main fault. In this scenario, the RSF model is too simplistic. This does not imply that the model is not worth developing, but a discussion of the possible real-world complications is needed.

We agree it is important to connect our findings to the real world. We have newly added the application of our model in a complex fault network, including the earthquake triggering between faults and fault segments and how it modifies the seismic hazard (line 718-725), and indicated the importance of the presence of fault network on other observations such as earthquake statistics (line 766-771).

Figure S4 NEEDS to be integrated in the main text to make it clearer and easier to read.

We added this figure to the main text (new Figure 2).

Line comments

14: I think the word “paradox” should be changed. As explained by other reviewers and by the authors themselves, it is fairly understood that velocity strengthening friction applies at steady conditions (during the nucleation) but does not prevent unstable fault slip in absolute terms).

We removed the word “paradox”.

19-20: I would be more cautious here. I suggest something like “subsequent slip on human lifetimes is likely to become increasingly quiet/stable”

We added clarification that our statement only holds for velocity-strengthening faults. Since this sentence describes our simulations, we kept the accurate term “stable” that only describes what we observed on VS faults. We don’t want to confuse the readers by phrasing our statement with uncertainty.

23-26: If you have the space in the abstract, you could specify HOW the site choice is affected by the insights provided by your work.

Due to the word limit we couldn’t add the site choice into the abstract. We do reformulate the abstract to better show the need for those professionals to read this paper, such that we hope they will read more to find this answer. We added a dedicated paragraph in the discussions for the site choice (line 740-765).

35: I would argue that the load is high but the strain rate is low, as it should be in plate interiors that “transmit” stress from the plate boundaries.

We replaced “tectonic loading rate” by “tectonic strain rate” (line 54).

70-71: I think this statement is broadly incorrect. VS may hinder slip nucleation, but may not prevent it (as also demonstrated by the other reviewer). For instance, when the nucleation patch is grown enough, the increasing frictional resistance behind the crack tip may not be enough to stop the nucleation. I would try to re-formulate this statement with a bit more nuance.

We deleted this first half of the sentence as suggested (line 92-95).

105: Also grain indentation by pressure-solution may contribute to cohesion and compaction.

We added the compaction via pressure solution as another healing mechanism (line 156, as used in Chen-Niemeijer-Spiers (CNS) model in Chen et al, 2019). We don't use grain indentation since it is a consequence of compaction via pressure solution.

104-106: the sentence is awkward to read. Maybe something like “cohesive rocks show evidence of cementation at the microscale”?

We rephased as suggested (line 159-161).

117-121: this is a rather strange statement. Healing is commonly known as the cause for the accumulation of new elastic energy for the seismic cycle. First is not entirely new that slow slip/creep has been linked to low healing only last year (e.g. Carpenter et al., 2011, Nature Geoscience; Tesei et al., 2012; JGR; and others, including some work from the Utrecht group!). Second, healing is inherent in RSF, as you said. Therefore, even if it was not explicitly applied to the specific case of induced seismicity, the RSF formulation should hold for all general cases, if you believe is a solid framework for fault friction modeling.

We removed this paragraph because it is not needed after we restructured the paper according to Nature Communication style guideline.

Nevertheless, we disagree with the Reviewer's comment. First, we have explained in a previous response (page 9) why healing is not important for natural seismicity so that it has been ignored in previous earthquake models.

Second and more importantly, we need to clarify that healing is not the cause for elastic energy accumulation. This is a common misunderstanding. Assume two rocks are held together without external loading: the interface gets strengthened but elastic energy does not accumulate simply because there is no loading (no elastic deformation). Shear loading, and hence elastic energy accumulation, is only accelerated during human activities (Fig. 9). In other words, faults are not becoming closer to failure due to frictional healing. This is an important message for non-experts as well. We added this clarification to a new paragraph in the discussion section (line 726-730).

317: Sometimes negative b is measured in the lab and, usually it is coupled with negative healing rates (albeit in short timescale-experiments). the possibility of negative b and effectively zero healing could be briefly considered with a couple of lines here.

We have conducted simulations with zero and negative b values in Fig. 8. We did not extend our discussion on this negative healing further, mainly because there is yet no physical explanation for this observation.

355: it is unclear here what “large enough” means here. Are you referring to a magnitude threshold, a minimum moment release, minimum slip, a simple earthquake that can break the surface...etc. Please specify.

We referred to the maximum slip rate, which acts as a proxy to seismic hazard. We replaced “large enough” by this specific matrix we investigated (line 522).

373-375: I would keep the original caveats (now deleted) that the analysis relies on the knowledge of frictional properties of the fault materials and that the last time of activity can be confidently estimated.

We added the information back (line 542-544).

405: 66 is not the correct reference for pointing the reader to carbonate-hosted earthquakes. Provide a couple of other citations.

We replaced (old) ref. 66 (Scuderi et al., 2016) with a series of observational works (Valoroso et al, Geology, 2014; Karanen et al., Science, 2014; Buijze et al., Netherlands Journal of Geosciences, 2019). We acknowledged the experimental works including Scuderi et al., 2016 in the second half of the sentence (line 584).

3.3 This whole paragraph about the nature of the “b” parameter is a bit speculative and does not add much to the paper. I suggest it can be safely deleted.

We deleted this section.

450: this is not surprising at all, since “b” is a fit parameter to describe the evolution of stress with slip or velocity. if it’s positive, it describes some form of slip weakening.

We deleted this section according to a previous comment.

460: also Pozzi et al., 2023 Geophysical Journal International <https://doi.org/10.1093/gji/ggad322>

We deleted this section according to a previous comment.

510-512: This statement is rather sharp. I suggest to rephrase it in a more dubitative way. either because faults contain heterogeneous material (both VS and VW/neutral), there are minor faults, and because your analysis has limitations that might not always make it applicable to nature.

To be accurate and more subtle, we used “VS fault segments” instead of “VS faults” and added the description on “VW fault segments” to cover the heterogeneous situations the Reviewer mentioned. As replied before, the discussions on heterogenous faults and fault networks are extended to make our results better applicable to nature (line 718-725, 766-771).